

# The Alongshore Tilt of Mean Dynamic Topography and its Implications for Model Validation and Ocean Monitoring

Christoph Renkl[1,2], Eric C.J. Oliver[2], and Keith R. Thompson[2,†]

[1]Physical Oceanography Department, Woods Hole Oceanographic Institution, Woods Hole, MA, USA
[2]Department of Oceanography, Dalhousie University, Halifax NS, Canada
[†]deceased

**Correspondence:** Christoph Renkl (christoph.renkl@whoi.edu)

**Abstract.** Mean dynamic topography (MDT) plays an important role in the dynamics of shelf circulation. Coastal tide gauge observations in combination with the latest generation of geoid models are providing estimates of the alongshore tilt of MDT with unprecedented accuracy. Additionally, high-resolution ocean models are providing better representations of nearshore circulation and the associated tilt of MDT along their coastal boundaries. It has been shown that the newly available geodetic

estimates can be used to validate model predictions of coastal MDT variability on global and basin scales. On smaller scales, however, there are significant variations in alongshore MDT that are on the same order of magnitude as the accuracy of the geoid models.

In this study, we use a regional ocean model of the Gulf of Maine and Scotian Shelf (GoMSS) to demonstrate that the new observations of geodetically referenced coastal sea level can provide valuable information also for the validation of such

high-resolution models. The predicted coastal MDT is in good agreement with coastal tide gauge observations referenced to the Canadian Gravimetric Geoid model (CGG2013a) including a significant tilt of alongshore MDT along the coast of Nova Scotia. Using the validated GoMSS model and several idealized models, we show that this alongshore tilt of MDT can be interpreted in two complementary, and dynamically consistent, ways: In the coastal view, the tilt of MDT along the coast can provide a direct estimate of the average alongshore current. In the regional view, the tilt can be used to approximate upwelling

averaged over an offshore area. This highlights the value of using geodetic MDT estimates for model validation and ocean monitoring.

## 1 Introduction

It has long been recognized that the alongshore tilt of mean dynamic topography (MDT) plays an important role in the dynamics of shelf circulation (e.g., Scott and Csanady, 1976; Csanady, 1978; Hickey and Pola, 1983; Werner and Hickey, 1983; Lentz,

2008). On the inner shelf (the region just outside the surf zone in water depths of order ten meters) frictional effects are dominant. Furthermore, due to the coastal constraint of no normal flow, currents in the nearshore mostly vary in alongshore direction (Lentz and Fewings, 2012). It has been shown that this results in continental shelves acting as a low wave number filter (Huthnance, 2004). This implies that mesoscale variations of sea level in the deep ocean are attenuated and only signals with large length scales on the order of thousand kilometers can be detected at the coast.



On the shelf, a multitude of drivers including wind stress, input of freshwater by rivers, and tidal rectification contribute to the circulation and thus impact the sea level at the coast (Lentz and Fewings, 2012). It is important to note that the coast acts as a waveguide and the effect of these drivers can be felt long distances "downstream" in the sense of coastal trapped wave propagation (e.g., Csanady, 1978; Thompson, 1986; Thompson and Mitchum, 2014; Frederikse et al., 2017; Hughes et al., 2019). The large number of drivers, and the possibility of remote effects, has resulted in debate about the origin of the observed

alongshore pressure gradient at the coast (e.g., Csanady, 1978; Chapman et al., 1986; Xu and Oey, 2011).

MDT appears in the momentum equation in the form of a gradient term and thus is an integrated measure of the mean circulation. This makes MDT a potentially useful variable for ocean monitoring and the validation of ocean models. The direct observation of MDT is complicated by the need to specify the geoid, relative to which MDT is defined. However, recent advances in geodesy have led to new and improved models of the geoid which can be used to get reliable estimates of MDT at

coastal tide gauges (Woodworth et al., 2012). These independent measurements are accurate on the centimeter level (Huang, 2017) and thus provide potentially valuable information for the validation of ocean and shelf circulation models.

Higginson et al. (2015) compared multiple global ocean models with geodetically referenced sea level observations along the east coast of North America using different geoid models. While they showed a general convergence between the estimates of MDT, they also pointed out that some models predicted a drop near Cape Hatteras that is not evident in the observations.

They concluded that these models did not capture the attenuation of the deep ocean signal over the shelf. A similar analysis was done by Lin et al. (2015) for the Pacific coasts of North America and Japan. They demonstrated a good agreement between the two approaches and furthermore used an analysis of the momentum budget along the coasts to illustrate the dominant dynamics behind the observed MDT. These studies as well as others (e.g., Hughes et al., 2015; Ophaug et al., 2015; Woodworth et al., 2015; Filmer et al., 2018) illustrate the value of the newly available geodetic estimates of coastal MDT for model validation.

On the other hand, the overall convergence of the geodetically estimated and predicted MDT simultaneously also increases confidence in the geoid models (Huang, 2017).

Most of the previous studies, including the ones mentioned above, focus on global and basin-scale variability of MDT at the coast. There are however significant variations on smaller scales that are on the same order of magnitude as the accuracy of the geoid models. In this study, we focus on the MDT along the northwest Atlantic coast predicted by the Gulf of Maine and Scotian

Shelf (GoMSS) model (Katavouta and Thompson, 2016). The circulation in this region is part of a large-scale buoyancy-driven coastal circulation originating along the south coast of Greenland (Chapman and Beardsley, 1989). On smaller scales, tidal rectification can generate mean currents up to $20\,\mathrm{cm\,s^{-1}}$ (Loder, 1980) and it has been shown that GoMSS is able to capture these processes well (Katavouta and Thompson, 2016; Katavouta et al., 2016).

This raises our first research question: Can new observations of geodetically referenced coastal sea level help validate high-

resolution regional ocean models? The second question is what can the alongshore tilt of MDT at the coast tell us about shelf circulation, followed by, what are the implications for coastal monitoring? These questions are of practical importance because (i) MDT provides an integrated measure of the mean circulation, (ii) tide gauges are cheap to deploy and maintain compared to many other oceanographic observing platforms (e.g., ships and gliders), and (iii) long records (several decades of hourly data) exist for some locations thereby providing background against which to interpret more recent variability. Using GoMSS and





several idealized models, we demonstrate that alongshore MDT can be used to estimate not only flow along the coast, but also area-integrated measures of upwelling offshore.

This study is structured as follows: Section 2 provides a description of the approaches to estimate coastal MDT from sea level observations and ocean models. In Section 3, two views of the dynamical role of the alongshore tilt of MDT at the coast are introduced. These views are illustrated in Section 4 using idealized models of ocean circulation. In Section 5, the mean

circulation and MDT predicted by GoMSS are presented and validated using geodetically referenced sea level measurements by tide gauges. The mean alongshore momentum balance predicted by GoMSS is discussed in Section 6. Section 7 provides an analysis of the predicted tilt of MDT along the coast of Nova Scotia with respect to the two dynamical interpretations. In Section 8, the results are summarized and implications for ocean monitoring are discussed.

## 2  Estimating the Alongshore Tilt of Coastal MDT

Mean dynamic topography (MDT), henceforth denoted by $\eta$, refers to the mean sea level (MSL) above the geoid corrected for the inverse barometer effect and averaged over a period of time to remove tidal and meteorological variations. MDT is also referred to as ocean dynamic sea level and is solely defined by ocean dynamics and density (Gregory et al., 2019). The alongshore tilt of MDT can be estimated using two independent approaches: a geodetic approach based on sea level observations and a hydrodynamic approach based on ocean circulation models. In this section, these approaches are outlined

and information about data used in this study is presented.

### 2.1  Geodetic Approach

In the geodetic approach, sea level measurements by tide gauges relative to tidal benchmarks are referenced to a common vertical datum which traditionally is estimated by spirit levelling (Huang, 2017). Recent advances by the geodetic community have led to new and improved high-resolution geoid models with an accuracy of several centimeters. These geoid models

provide the geoid height relative to a reference ellipsoid. Through satellite-based navigation systems (e.g., Global Positioning System, GPS), sea level heights measured by tide gauges relative to the same ellipsoid can be determined. Subtracting the local geoid height yields an estimate of the MDT

$$\eta = \eta_{\text{BM}} + h_{\text{e}} - N, \tag{1}$$

where $\eta_{\text{BM}}$ is the MSL relative to the GPS tidal benchmark with height $h_{\text{e}}$ above the reference ellipsoid, and $N$ is the geoid

height above the same ellipsoid.

MSL values were computed from hourly observations of sea level at available tide gauges in the Gulf of Maine and Scotian Shelf area for the period 2011–2013. These tide gauges measure the real, observed height of the air-sea interface using acoustic, microwave radar, or air pressure-compensated pressure sensors. Since the focus of this study is on the regional-scale MDT signal, only tide gauges that are not influenced by highly localized effects were considered (see below). Table 1 gives a summary



of the stations used in this study and their locations are shown in Figure 1. Overall, the proportion of missing values over the study period is less than 3% at all stations.

For stations in the USA, hourly water level records with respect to Mean Lower Low Water (MLLW) were retrieved from the National Oceanic and Atmospheric Administration (NOAA). The tide gauge in Chatham, Lydia Cove, MA (NOAA ID #8447435) was excluded because of its location in a shallow lagoon behind a series of sandbars. GPS ellipsoidal heights at

nearby benchmarks were obtained from the Online Positioning User Service (OPUS) provided by the National Geodetic Survey (NGS). Their shared solutions list benchmark coordinates relative to the North American Datum, NAD83(2011) epoch 2010.0. They were converted to the International Terrestrial Reference System, ITRF2008 epoch 2010.0 (Altamimi et al., 2011), using the Horizontal Time-Dependent Positioning tool (HTDP, Pearson and Snay, 2013) provided by NGS. At benchmarks where multiple OPUS shared solutions were available, the one with the smallest uncertainty in observed ellipsoidal height was chosen.

Using information from benchmark sheets about the relative height of the benchmarks with respect to MLLW, the sea level observations were expressed relative to the GRS80 ellipsoid.

For tide gauges in Canada, hourly water level records with respect to chart datum (CD) were obtained from the Canadian Hydrographic Service (CHS). GPS ellipsoidal heights were obtained for nearby benchmarks of the Natural Resources Canada (NRCAN) High Precision 3D Geodetic Network in the ITRF2008 epoch 2010.0 reference frame. Generally, the height of the

benchmark relative to CD is not known, but can be inferred from orthometric height differences with tidal benchmarks of NRCAN's Vertical Passive Control Network published by CHS. Using this information, MSL can be expressed relative to the GRS80 ellipsoid.

Note that the tide gauge for Saint John, NB, Canada (CHS ID #65) was excluded because it is situated in the mouth of St. John River and sheltered by breakwaters. It follows that sea level variations at this tide gauge are likely to be dominated by

local processes (e.g., tides and river discharge). The permanent tide gauges located in Halifax, NS, Canada (CHS ID #490) and at the Bedford Insitute of Oceanography, Dartmouth, NS, Canada (CHS ID #491) are only a few kilometers apart. Here, the record at the latter will be used because it has fewer missing values and is closer to the GPS benchmark. (The resulting MDTs agree within millimeters.)

Geodetic estimates of coastal MDT were then computed using (1). Here, the Canadian Gravimetric Geoid model of 2013 -

Version A (CGG2013a, Véronneau and Huang, 2016) was used to provide the geoid height $N$ relative to the GRS80 ellipsoid in the ITRF2008 epoch 2010.0 reference frame as well as a measure of its accuracy. The CGG2013a geoid heights are available on a grid with $2'$ spacing. These were bilinearly interpolated to the benchmark locations and then subtracted from the MSL referenced to the benchmarks.

Uncertainties in the geodetic MDT estimates for the study period arise from errors in the GPS ellipsoidal heights as well

as geoid height. These uncertainties are independent and their standard deviations are known. It was therefore possible to use conventional error propagation rules to estimate the standard error of the geodetically determined MDT. The main source of uncertainty is the estimated error in the CGG2013a geoid height which is generally one order of magnitude higher compared to errors in the ellipsoidal heights. Overall, the uncertainties in MDT are typically less than $1.6\,\mathrm{cm}$ (Table 1).





GPS coordinates are generally expressed in a tide-free coordinate system (Woodworth et al., 2012) as is the geoid model
CGG2013a. In order to make geodetically referenced MSL observations comparable to ocean circulation models, mean tidal
effects on the coordinate systems have to be considered. Following Ekman (1989), the geodetic MDT estimates were converted
from tide-free to mean tide coordinates. Note that the minus sign error reported by Woodworth et al. (2012) was taken into
account.

Since MDT is solely defined by ocean dynamics and density (Gregory et al., 2019), the geodetic MDT estimates were
corrected for the inverse barometer effect following Andersen and Scharroo (2011). Here, 6-hourly data of air pressure reduced
to MSL from the NCEP Climate Forecast System Version 2 (CFSv2, Saha et al., 2014) were used. The time-mean air pressure
$p_a$ at the grid point closest to the tide gauges was used to compute the mean inverse barometer correction in centimeters

$$\eta_{\mathrm{IB}} = \frac{p_a - p_{\mathrm{ref}}}{\rho_0 g} = 0.99485 \, \mathrm{cm\,hPa}^{-1} \left( p_a - p_{\mathrm{ref}} \right), \tag{2}$$

which was added to the geodetic MDT estimates. Here, $p_{\mathrm{ref}} = 1013.0 \, \mathrm{hPa}$ is the atmospheric reference pressure. The difference
in the mean inverse barometer effect between the tide gauges in Boston and North Sydney is $2 \, \mathrm{cm}$.

## 2.2    Hydrodynamic Approach

Ocean circulation models typically have their vertical coordinate system expressed relative to an equipotential surface assumed
to be the geoid. Therefore, MSL predicted by the model is equal to the MDT and can be directly compared to the geodetic
estimates. This is referred to as the hydrodynamic or ocean approach (e.g., Woodworth et al., 2012).

To estimate the MDT along the northwest Atlantic coast, we use the Gulf of Maine and Scotian Shelf (GoMSS, Figure 1)
model developed by Katavouta and Thompson (2016) and upgraded by Renkl and Thompson (2022). GoMSS is based on
version 3.6 of the Nucleus for European Modelling of the Ocean (NEMO, Madec et al., 2017). In comparison to the original
configuration, the bathymetry was replaced with a combination of the $30''$ GEBCO bathymetry (Weatherall et al., 2015) and
high-resolution in-situ measurements using an optimal interpolation procedure. This was done to ensure the bathymetry is
accurately represented in GoMSS, particularly in shallow regions. GoMSS has a horizontal grid spacing of 1/36° corresponding
to 2.1 to 3.6 $\mathrm{km}$ in the study region. The model has 52 vertical levels that increase in thickness from 0.72 m to 235.33 m in
a state of rest, but vary in time via a variable volume formulation of the nonlinear free surface (z*-coordinate, Levier et al.,
2007). Partial cells at the bottom ensure a better resolution of the bathymetry that is clipped at 4000 m.

Both the initial conditions and lateral boundary forcing are based on water temperature, salinity, sea surface height, and
currents from the GLORYS12v1 reanalysis (Lellouche et al., 2021). Additionally, tidal elevation and currents for five tidal
constituents ($M_2$, $N_2$, $S_2$, $K_1$, $O_1$) from FES2004 (Lyard et al., 2006) were prescribed along the lateral boundaries. Atmospheric
forcing at the air-sea boundary was based on the CFSv2 (Saha et al., 2014).

The following analysis is based on daily mean output fields of a hindcast for the period 2011–2013. Note that GoMSS does
not include forcing by atmospheric pressure and therefore no corrections for the inverse barometer effect have to be applied to
the model output.



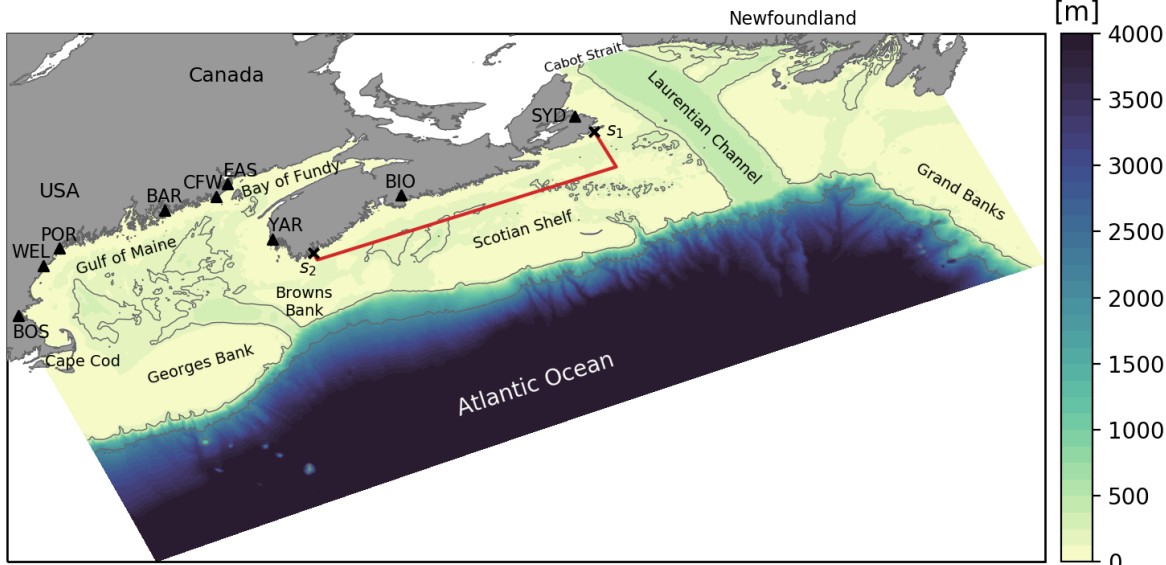

**Figure 1.** GoMSS model domain and tide gauge locations for the Scotian Shelf, Gulf of Maine and Bay of Fundy. Contours indicate the 200 and 2000 m isobaths. The triangles indicate the tide gauge locations with the abbreviations referring to the stations listed in Table 1. The area enclosed by the red polygon and the coastline illustrates the region over which the regional view is evaluated and the markers $s_1$ and $s_2$ indicate reference points along the coast.

Model predictions of the alongshore MDT from the hydrodynamic approach are based on the predicted MSL over the three-year period. The coastal MDT is taken at the wet (non-land) grid cell closest to the coast and the alongshore tilt of MDT, in the following denoted by $\Delta\eta_c$, is the difference in MDT between two points along the coast.

## 3  Dynamical Interpretation of $\Delta\eta_c$

In the steady limit, the depth-averaged momentum equation can be written

$$g\nabla\left(\eta - \eta_s + \frac{|\overline{\mathbf{u}}|^2}{2g} + \frac{p_a}{g\rho_0}\right) = -(f+\zeta)\hat{\mathbf{k}} \times \overline{\mathbf{u}} - \frac{1}{h}\nabla\chi + \frac{\boldsymbol{\tau}^w - \boldsymbol{\tau}^b}{h} + \overline{\mathbf{D}}_1, \tag{3}$$

where

$$h = H + \eta \tag{4}$$

is the total water depth. On the left-hand side of (3), all gradient terms including the steric contribution to sea level ($\eta_s$), the Bernoulli setdown due to the depth-averaged current ($\overline{\mathbf{u}}$), and the inverse Barometer effect have been combined and multiplied by the vertical acceleration due to gravity $g$. Therefore, (3) can be considered an equation for the gradient of the dynamically





active component of sea level that is balanced by the Coriolis term with parameter $f$ modified to include the effect of relative vorticity $\zeta$, the gradient of the depth-integrated potential energy anomaly ($\chi$), the difference between wind stress ($\boldsymbol{\tau}^{\mathrm{w}}$) and bottom stress ($\boldsymbol{\tau}^{\mathrm{b}}$), and lateral mixing ($\overline{\mathbf{D}}_{\mathrm{l}}$).

Taking the curl of (3), leads to the following equation for the vorticity of the depth-averaged flow:

$$\overline{\mathbf{u}} \cdot \nabla (f+\zeta) - \frac{(f+\zeta)}{h} \overline{\mathbf{u}} \cdot \nabla h = J\left(\chi, h^{-1}\right) + \hat{\mathbf{k}} \cdot \nabla \times \left(\frac{\boldsymbol{\tau}^{\mathrm{w}} - \boldsymbol{\tau}^{\mathrm{b}}}{h}\right) + \hat{\mathbf{k}} \cdot \nabla \times \overline{\mathbf{D}}_{\mathrm{l}}, \tag{5}$$

Using the fact that $\frac{\partial}{\partial t}(f+\zeta) = 0$ for steady flow, the left-hand side can be summarized as

$$h \frac{D}{Dt}\left(\frac{f+\zeta}{h}\right) = \frac{\partial}{\partial t}(f+\zeta) + \overline{\mathbf{u}} \cdot \nabla (f+\zeta) - \frac{(f+\zeta)}{h} \overline{\mathbf{u}} \cdot \nabla h \tag{6}$$

describing the total change of potential vorticity following a fluid column due to the local rate of change and advection of

absolute vorticity as well as vortex tube stretching by flow across isobaths.

The first term on the right-hand side of (5) is the joint effect of baroclinicity and relief (JEBAR, Sarkisyan and Ivanov, 1971)

$$J\left(\chi, h^{-1}\right) = \frac{\partial \chi}{\partial x} \frac{\partial h^{-1}}{\partial y} - \frac{\partial \chi}{\partial y} \frac{\partial h^{-1}}{\partial x}. \tag{7}$$

As shown by Mertz and Wright (1992), this can be interpreted as the curl of a horizontal force exerted on the fluid by the bottom.

They furthermore showed that JEBAR acts as a correction to the vortex stretching term in (5) by removing the nonphysical contribution of the geostrophic flow referenced to the bottom. Note that JEBAR vanishes in the case of constant water depth. The remaining terms in (5) describe the net torque exerted by the difference between wind stress and bottom friction, and vorticity dissipation due to lateral mixing.

In the following, (3) and (5) will be used to explore the role of $\Delta\eta_{\mathrm{c}}$ in coastal and shelf circulation.

## 3.1 Interpretation of $\Delta\eta_{\mathrm{c}}$ in Terms of Coastal Circulation

At the coast, the depth-averaged momentum balance (3) simplifies. Due to the condition of no flow across the coastal boundary, the first term on the right-hand side vanishes in the alongshore component. Furthermore, it is assumed that density variations along the coast, i.e., the combined effect of $\nabla\eta_{\mathrm{s}}$ and $\frac{1}{h}\nabla\chi$, can be neglected. This assumption will be shown to be reasonable in the analysis of the alongshore momentum balance predicted by GoMSS in Section 6. It follows from (3), under the assumption

of steady state, that the momentum equation in *alongshore direction* reduces to

$$g\nabla\left(\eta + \frac{p_{\mathrm{a}}}{g\rho_0} + \frac{|\overline{\mathbf{u}}|^2}{2g}\right) = \frac{\boldsymbol{\tau}^{\mathrm{w}} - \boldsymbol{\tau}^{\mathrm{b}}}{h} + \overline{\mathbf{D}}_{\mathrm{l}}. \tag{8}$$

Given the inverse barometer effect has been removed from the observations and is not included in GoMSS, the atmospheric pressure term in (8) will be ignored. The left-hand side is the gradient of sea level corrected for the Bernoulli effect. The Bernoulli term is typically only significant where changes in current speed occur over small distances, e.g., around headlands

and in channels (e.g., Renkl and Thompson, 2022). Thus, the large-scale alongshore gradient of MDT at the coast, is primarily balanced by the sum of wind stress, bottom drag, and horizontal mixing.



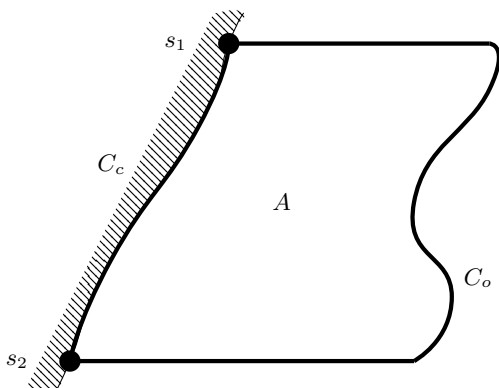

**Figure 2.** Schematic of closed curve along which the momentum balance is integrated. The hatched area is the land and the bold line illustrates the closed integration path $C$ which can be divided into a coastal ($C_{\mathrm{c}}$) and offshore part ($C_{\mathrm{o}}$). The area enclosed by $C$ is denoted by $A$.

Suppose for now that the Bernoulli term can be neglected. In that case, the integral of (8) along a curve $C_{\mathrm{c}}$ following the coastline between two points (see Figure 2) gives the large-scale alongshore balance of the tilt of MDT in vector form

$$g\Delta\eta_{\mathrm{c}} = \int_{C_{\mathrm{c}}} \left[ \frac{\boldsymbol{\tau}^{\mathrm{w}} - \boldsymbol{\tau}^{\mathrm{b}}}{h} + \overline{\mathbf{D}}_{\mathrm{l}} \right] \cdot d\mathbf{r}. \tag{9}$$

This is one interpretation of $\Delta\eta_{\mathrm{c}}$ in terms of coastal circulation. From (9) it is clear that, along the coast, the tilt of MDT is in frictional equilibrium. In the following, this interpretation is referred to as the *coastal view*.

In the special case when the wind setup along the coast

$$g\Delta\eta_{\mathrm{w}} = \int_{C_{\mathrm{c}}} \frac{\boldsymbol{\tau}^{\mathrm{w}}}{h} \cdot d\mathbf{r} \tag{10}$$

is known, a new variable $\tilde{\eta}$ can be defined as the wind-corrected MDT. More generally, $\tilde{\eta}$ can also incorporate corrections for the Bernoulli effect and atmospheric pressure variations as we have already done. Thus, (9) becomes

$$g\Delta\tilde{\eta}_{\mathrm{c}} = -\int_{C_{\mathrm{c}}} \left[ \frac{\boldsymbol{\tau}^{\mathrm{b}}}{h} - \overline{\mathbf{D}}_{\mathrm{l}} \right] \cdot d\mathbf{r}. \tag{11}$$

If $\boldsymbol{\tau}^{\mathrm{b}}$ is parameterized in terms of the depth-averaged current, $\overline{\mathbf{D}}_{\mathrm{l}}$ can be neglected (e.g., outside a narrow viscous boundary layer), and the wind setup along the coast is known, it will be shown that $\Delta\tilde{\eta}_{\mathrm{c}}$ can be interpreted as a measure of the average alongshore flow between two points along the coast.

## 3.2 Interpretation of $\Delta\eta_{\mathrm{c}}$ in Terms of Regional Circulation

Instead of integrating the momentum balance along the coast, it is also possible to define an offshore curve $C_{\mathrm{o}}$ from $s_2$ to $s_1$ along which (3) can be integrated. Together with the coastal integration path, this forms a closed curve $C = C_{\mathrm{c}} + C_{\mathrm{o}}$ (Figure



2). Note that the closed line integral of the sea level gradient term along $C$ is zero and so

$$\int_{C_c} \nabla\eta \cdot d\mathbf{r} + \int_{C_o} \nabla\eta \cdot d\mathbf{r} = 0. \tag{12}$$

This demonstrates that the tilt of MDT along the coast $\Delta\eta_c$ must equal the drop along the offshore integration path $\Delta\eta_o$:

$$\Delta\eta_c = -\Delta\eta_o.$$

Since $\Delta\eta_o$ is determined by the regional ocean dynamics, it follows that $\Delta\eta_c$ can also be interpreted in terms of the regional ocean dynamics.

Using Green's theorem (Green, 1828), the line integral of a two-dimensional vector field $\mathbf{F}$ along a closed curve $C$ is equal

to the surface integral of the curl of the field over the enclosed area $A$,

$$\oint_C \mathbf{F} \cdot d\mathbf{r} = \iint_A (\nabla \times \mathbf{F}) \cdot \hat{\mathbf{n}}\, dA, \tag{13}$$

where $\hat{\mathbf{n}}$ is the unit vector perpendicular to the surface $A$.

For the case of depth-averaged ocean circulation, the term under the area integral is the relative vorticity of the flow field. Therefore, the circuit integral of the momentum equation is equal to the area integral of the vorticity equation. Combining (3)

and (13) with the vorticity equation (5) gives

$$\oint_C \left[ \frac{\boldsymbol{\tau}^w - \boldsymbol{\tau}^b}{h} + \overline{\mathbf{D}}_1 \right] \cdot d\mathbf{r} = \iint_A \left[ h\frac{D}{Dt}\left( \frac{f+\zeta}{h} \right) - J\left(\chi, h^{-1}\right) \right] dA. \tag{14}$$

The left-hand side is the closed line integral of the frictional terms which is balanced by the area integral of total change of potential vorticity and the JEBAR term. Note that all the gradient terms, including the sea level gradient, have dropped out.

The circuit integral on the left-hand side can be split into a coastal and offshore part. Note that the line integral along the

coast is equal to $g\Delta\eta_c$. Hence, substituting (9) in (14) gives

$$g\Delta\eta_c = \iint_A \left[ h\frac{D}{Dt}\left( \frac{f+\zeta}{h} \right) - J\left(\chi, h^{-1}\right) \right] dA - \int_{C_o} \left[ \frac{\boldsymbol{\tau}^w - \boldsymbol{\tau}^b}{h} + \overline{\mathbf{D}}_1 \right] \cdot d\mathbf{r}, \tag{15}$$

where the second term on the right-hand side is the offshore segment of the circuit integral in (14). This is another interpretation of $\Delta\eta_c$, this time in terms of regional ocean dynamics. It shows that the alongshore tilt of MDT at the coast can also be interpreted as an integrated measure of the regional ocean circulation. In the following this interpretation will be referred to as

the *regional view*.

Both the coastal and regional views of $\Delta\eta_c$ are complementary and dynamically consistent: the offshore circulation drives the coastal dynamics, and on the other hand, the dynamics along the coast act as a boundary condition for the offshore circulation.

A case of special interest assumes steady state, small Rossby number, constant Coriolis parameter $f$, and $\eta \ll H$. Under these assumptions, the vortex stretching term in (15) is given by the depth-averaged flow across isobaths:

$$h\frac{D}{Dt}\left( \frac{f+\zeta}{h} \right) = -\frac{f}{H}\overline{\mathbf{u}} \cdot \nabla H. \tag{16}$$





Consider now the JEBAR term, $J\left(\chi, H^{-1}\right)$. Mertz and Wright (1992) showed

$$J(\chi, H^{-1}) = -\frac{f}{H}(\overline{\mathbf{u}}_{\mathrm{g}} - \mathbf{u}_{\mathrm{g,b}}) \cdot \nabla H, \tag{17}$$

where $\mathbf{u}_{\mathrm{g}}(z)$ is a geostrophically balanced horizontal velocity defined in terms of the density field according to the following thermal wind equation

$$f\hat{\mathbf{k}} \times \frac{\partial \mathbf{u}_{\mathrm{g}}}{\partial z} = g\nabla \epsilon \tag{18}$$

with bottom boundary condition $\mathbf{u}_{\mathrm{g}}(-H) = \mathbf{u}_{\mathrm{g,b}}$. Here, $\epsilon = (\rho - \rho_0)/\rho_0$ is the normalized density perturbation. Mertz and Wright (1992) used (17) to show the JEBAR term *"represents precisely the geostrophic component of the correction to the topographic stretching term to account for the fact that the bottom velocity, not the depth-averaged velocity, yields topographic vortex-tube stretching"*.

Combining (16) and (17) gives

$$h\frac{D}{Dt}\left(\frac{f+\zeta}{h}\right) - J\left(\chi, h^{-1}\right) = -\frac{f}{H}\mathbf{u}^* \cdot \nabla H, \tag{19}$$

where

$$\mathbf{u}^* = \overline{\mathbf{u}} - \overline{\mathbf{u}}_{\mathrm{g}} + \mathbf{u}_{\mathrm{g,b}} \tag{20}$$

can be interpreted as the depth-averaged current corrected for the geostrophic current at the bottom. The upwelling velocity caused by $\mathbf{u}^*$ running across a sloping seafloor is

$$w^* = -\mathbf{u}^* \cdot \nabla H. \tag{21}$$

With this definition of $w^*$, the regional view expressed by (15) can be written as

$$g\Delta\eta_{\mathrm{c}} \approx f\iint\limits_A \frac{w^*}{H}\, dA - \int\limits_{C_{\mathrm{o}}}\left[\frac{\boldsymbol{\tau}^{\mathrm{w}} - \boldsymbol{\tau}^{\mathrm{b}}}{h} + \overline{\mathbf{D}}_{\mathrm{l}}\right] \cdot d\mathbf{r}. \tag{22}$$

In the following sections, several idealized ocean models are used to dynamically interpret $\Delta\eta_{\mathrm{c}}$ and illustrate the potential of the tilt of MDT for ocean monitoring.

## 4 $\Delta\eta_{\mathrm{c}}$ in Idealized Ocean Models

In this section the coastal and regional views of $\Delta\eta_{\mathrm{c}}$ are illustrated using idealized ocean models. First, the model of a wind-driven basin circulation of Stommel (1948) is discussed. Second, two conceptual models of coastally trapped shelf circulation based on Csanady (1982) are illustrated. These models highlight the usefulness of $\Delta\eta_{\mathrm{c}}$ for the validation of ocean models.





### 4.1 Wind-Driven Gyre

The seminal model of Stommel (1948) describes the steady wind-driven circulation in an idealized, rectangular ocean basin on a $\beta$-plane (i.e., Coriolis parameter $f$ is a function of latitude) with dimensions $L_x$ and $L_y$ in $x$- (zonal) and $y$-direction (meridional), respectively. The water depth $H$ is assumed to be constant. Variations in atmospheric pressure are ignored and advection, density variations as well as lateral mixing are neglected.

Under these assumptions, the depth-averaged momentum equation (3) becomes

$$g\nabla\eta = f\hat{\mathbf{k}} \times \overline{\mathbf{u}} + \frac{\boldsymbol{\tau}^{\mathrm{w}} - \boldsymbol{\tau}^{\mathrm{b}}}{H}, \tag{23}$$

where the Coriolis parameter $f = f_0 + \beta y$ is a linear function of latitude. Bottom friction is assumed to be a linear function of the depth-averaged current, that is $\boldsymbol{\tau}^{\mathrm{b}} = \lambda\overline{\mathbf{u}}$, where $\lambda$ is the bottom friction coefficient with units $\mathrm{m\,s^{-1}}$. The wind forcing is taken to be purely zonal and prescribed as a sinusoidal function of latitude:

$$\tau_x^{\mathrm{w}} = -F\cos\left(\frac{\pi y}{L_y}\right) \quad \text{and} \quad \tau_y^{\mathrm{w}} = 0, \tag{24}$$

where $\tau_x^{\mathrm{w}}$ and $\tau_y^{\mathrm{w}}$ are the zonal and meridional components, respectively, of $\boldsymbol{\tau}^{\mathrm{w}}$ and $F$ is the maximum amplitude of the wind stress (see grey arrows in Figure 3).

Multiplying by $H$ and taking the curl of (23) yields the vorticity equation of the Stommel (1948) model

$$\beta V = -\frac{\partial \tau_x^{\mathrm{w}}}{\partial y} - \frac{\lambda}{H}\left(\frac{\partial V}{\partial x} - \frac{\partial U}{\partial y}\right), \tag{25}$$

where $U = \overline{u}H$ and $V = \overline{v}H$ are the volume transports in zonal and meridional direction, respectively. Introducing a stream function $\psi$, the components of the volume transport vector can be written as

$$U = \frac{\partial \psi}{\partial y} \quad \text{and} \quad V = -\frac{\partial \psi}{\partial x}. \tag{26}$$

Substituting these expressions in (25) gives

$$\nabla^2\psi + \frac{H\beta}{\lambda}\frac{\partial \psi}{\partial x} = \frac{F\pi}{\lambda L_y}\sin\left(\frac{\pi y}{L_y}\right), \tag{27}$$

which can be integrated to obtain a solution for the stream function.

Figure 3a shows the spatial structure of $\psi$ and the associated sea level for the Stommel model. Based on the Sverdrup relation (Sverdrup, 1947), the curl of the wind stress leads to convergence of the Ekman transport in the surface layer. As a result, downwelling (vortex squashing) occurs causing overall southward transport in the ocean interior to conserve potential vorticity. This southward transport is balanced by a swift and narrow current along the western boundary. Along this boundary, the model predicts a sea level tilt of 87 cm (Figure 3b).

Let the integration path $C$ be defined along the domain boundaries where the Coriolis term is zero because of the no-flow coastal boundary condition. Based on the assumptions above, the coastal view in (9) for the Stommel model becomes

$$\Delta\eta_{\mathrm{c}} = \frac{\lambda}{gH}\int\limits_0^{L_y}\overline{v}(0,y)\,dy. \tag{28}$$





**Figure 3.** Stommel (1948) model of wind-driven circulation. (a) Transport stream function (contours) illustrating an overall southward transport in the interior and a narrow return flow along the western boundary. Grey arrow show the wind stress forcing with maximum value $F = 0.1\,\mathrm{N\,m^{-2}}$. (b) Sea surface height including $\Delta\eta_c = 87\,\mathrm{cm}$ along the western boundary which is balanced by bottom friction $\tau^b_y = \lambda\overline{v}$ with $\lambda = 0.02\,\mathrm{m\,s^{-1}}$.





This shows that $\Delta\eta_c$ is a measure of the mean alongshore current.

Similarly, the regional view in (15), applied over the whole model domain, reduces to

$$\Delta\eta_c = \frac{L_x}{gH}\left[\tau^w(L_y) - \tau^w(0)\right]. \tag{29}$$

Note that the Sverdrup transport in the ocean interior is balanced by the volume transport in the western boundary current and therefore, its area integral over the whole domain is zero. From (29) it is clear that $\Delta\eta_c$ is also equal to the sea level setup due to the wind along the northern and southern boundary and hence a measure of the circulation offshore. It is important to point

out that in this interpretation, $\Delta\eta_c$ only depends on the wind stress and basin dimensions, but is independent of the bottom friction coefficient.

The right-hand side of (29) is equal to the area-integrated curl of the wind stress which is directly related to the Ekman pumping velocity $w_E$ at the base of the surface Ekman layer (e.g., Gill, 1982)

$$w_E = \hat{\mathbf{k}} \cdot \nabla \times \left(\frac{\boldsymbol{\tau}^w}{f}\right). \tag{30}$$

Hence, the regional interpretation of $\Delta\eta_c$ can be written as

$$\Delta\eta_c = \frac{1}{gH}\iint_A f w_E \, dx \, dy. \tag{31}$$

This demonstrates that, from a regional perspective, the tilt of MDT along the coast is a measure of the net surface downwelling over the whole basin.

Instead of applying the regional view in (15) over the entire model domain, it is also possible to define $C$ such that $C_o$ is

along the outer edge of the western boundary current, where $V = 0$. Integrating the vorticity equation (25) with respect to $x$ and substituting the alongshore momentum equation gives

$$g\frac{\partial\eta}{\partial y}\bigg|_{x=0} = -\frac{\beta}{gH}\int_0^L V \, dx. \tag{32}$$

This shows that, the alongshore gradient of MDT is also a measure of volume transport in the western boundary current. As shown by Stewart (1989), this idea can also be extended to inertial boundary currents.

By continuity, the volume transport in the western boundary current is equal to the southward Sverdrup transport in the interior of the domain. This in turn is proportional to the wind stress curl and also the overall downwelling in the model. This shows that all interpretations of the alongshore tilt of MDT are physically consistent.

### 4.2  Coastally Trapped Circulation

The role of the alongshore tilt of MDT in the circulation on continental shelves can be illustrated with the conceptual models

discussed by Csanady (1982) that focus on flow trapped within the coastal boundary layer. Consider a coordinate system where the $y$-axis is aligned with a straight coastline and the $x$-axis pointing offshore. Without lateral mixing and assuming the flow





to be steady, linear, and barotropic, the governing equations (3) can then be written in component form as

$$g\frac{\partial \eta}{\partial x} = \frac{f}{H}V + \frac{\tau_x^{\mathrm{w}}}{H},$$
(33)

$$g\frac{\partial \eta}{\partial y} = -\frac{f}{H}U + \frac{\tau_y^{\mathrm{w}} - \tau_y^{\mathrm{b}}}{H},$$
(34)

where $U$ and $V$ are the $x$- and $y$-components of the transport vector $\mathbf{U} = \overline{\mathbf{u}}H$. The Coriolis parameter $f$ is assumed to be constant and bottom friction is taken to be linearly proportional to the depth-averaged alongshore current, $\tau_y^{\mathrm{b}} = \lambda\overline{v}$. Under the long-wave approximation that the alongshore current is much larger than the cross-shore current, the bottom friction in $x$-direction can be neglected.

Cross-differentiating (33) and (34) yields the vorticity equation of this model

$$f\frac{\overline{u}}{H}\frac{\partial H}{\partial x} = -\frac{\partial}{\partial x}\left(\frac{\tau_y^{\mathrm{w}}}{H}\right) + \frac{\partial}{\partial y}\left(\frac{\tau_x^{\mathrm{w}}}{H}\right) + \frac{\partial}{\partial x}\left(\frac{\tau_y^{\mathrm{b}}}{H}\right).$$
(35)

The net torque exerted by the wind stress and bottom drag (right-hand side), is balanced by vortex stretching/squashing through movement into deeper or shallower water, respectively. This flow across isobaths results in convergence or divergence near the seafloor leading to bottom stress-induced Ekman pumping.

Equation (34) can be rearranged to get an expression for $\overline{u}$ which can be substituted in (35). Parameterizing bottom friction

with the alongshore geostrophic current times a drag coefficient $\lambda$ yields a single governing equation for the sea level

$$\frac{\partial^2 \eta}{\partial x^2} + \frac{f}{\lambda}\frac{\partial H}{\partial x}\frac{\partial \eta}{\partial y} = \frac{f}{g\lambda}\left(\frac{\partial \tau_y^{\mathrm{w}}}{\partial x} - \frac{\partial \tau_x^{\mathrm{w}}}{\partial y}\right).$$
(36)

Csanady (1982) pointed out the similarity of (36) to the heat conduction equation with downstream direction $-y$ corresponding to time. He furthermore used this analogy to discuss coastally trapped flow fields with respect to different forcing. In the following, two cases will be explored and the role of $\Delta\eta_{\mathrm{c}}$ discussed.

### 4.2.1   Wind Stress Along Portion of Coast

Assume water depth increases linearly with distance from shore as $H(x) = H_0 + sx$ where $s$ is a constant slope. The wind stress along the part of the domain where $0 \leq y \leq Y$ is taken to be constant and in alongshore direction only, $\boldsymbol{\tau}^{\mathrm{w}} = (0, \tau_y^{\mathrm{w}})$.

As shown above the dashed line in Figure 4, this wind stress causes an Ekman transport toward the coast. From (35) it can be seen that the wind stress over the sloping shelf as well as the flow across isobaths into shallower water exert a negative torque

on the water column. Thus, the flow is steered to the left resulting in an alongshore current at the coast in the direction of the wind. Consequently, sea level piles up in the downstream direction.

Applying the assumptions above to (9), the coastal view of $\Delta\eta_{\mathrm{c}}$ becomes

$$\Delta\eta_{\mathrm{c}} = \frac{\tau_y^{\mathrm{w}}Y}{gH_0} - \frac{\lambda}{gH_0}\int_0^Y \overline{v}(0,y)\,dy.$$
(37)





**Figure 4.** Stream function and sea surface height for two models of coastally trapped circulation. Water depth is increasing in $x$-direction as $H(x) = sx$, the bottom friction coefficient is $\lambda = 0.5 \times 10^{-3}\,\mathrm{m\,s^{-1}}$, and the Coriolis parameter $f = 10^{-3}\,\mathrm{s^{-1}}$. For $y > 0$, a spatially uniform wind stress $\tau_y^{\mathrm{w}} = u_*^2 = -0.01\,\mathrm{m\,s^{-2}}$ (grey arrows) is applied (adapted from Csanady, 1982).





The first term on the right-hand side is the wind setup along the coast. As expected, $\Delta\eta_c$ is in frictional equilibrium and
balances the difference between wind stress and bottom drag. If the wind setup along the coast is known, the corrected tilt of
MDT along the coast $\Delta\tilde{\eta}_c$ can be used as a direct measure of the mean alongshore current.

The regional view can be directly obtained from (22) under the assumption of barotropic flow which implies $\mathbf{u}^* = \overline{\mathbf{u}}$. If the
offshore integration path is chosen to be in deep water where the wind stress and bottom friction terms are negligible due to
their inverse dependence on $H$, (22) becomes

$$\Delta\eta_c = -\frac{fs}{g} \int\limits_0^Y \int\limits_0^{L_x} \frac{\overline{u}}{H} \, dx \, dy. \tag{38}$$

This shows that, from a regional perspective, $\Delta\eta_c$ is equal to the cross-shore Ekman transport due to the wind forcing and the
associated flow across isobaths. This onshore flow implies an overall upwelling in the area that can be monitored by observing
the sea level at the coast.

### 4.2.2    Coastal Mound

Assume that the wind stress vanishes for $y < 0$ and the flow field is established by prescribing a cross-shore sea level distribu-
tion $\eta = \eta_0(x)$ at $y = 0$ which is the result of some upstream process e.g., wind-driven onshore transport as discussed above. It
can be seen from (33) that the corresponding alongshore current is in geostrophic balance.

Figure 4 shows the resulting stream function and the associated sea surface height. The streamlines indicate a predominantly
alongshore flow, but they also show a spreading in offshore direction further downstream. Equation (35) shows that this cross-
shore flow is caused by the frictional torque at the sea floor acting on the alongshore current.

From (9) and (22), this offshore flow across isobaths can be directly related to the alongshore tilt of MDT at the coast

$$\Delta\eta_c = \underbrace{\frac{\lambda}{gH_0} \int\limits_{-y}^0 \overline{v}(0,y) \, dy}_{\text{coastal}} = \underbrace{-\frac{fs}{g} \int\limits_{-y}^0 \int\limits_0^{L_x} \frac{\overline{u}}{H} \, dx \, dy}_{\text{regional}}. \tag{39}$$

This shows again that, from a coastal point of view, $\Delta\eta_c$ is proportional to the mean alongshore current driven by the pressure
gradient. In the regional interpretation, $\Delta\eta_c$ is a measure of the area-integrated vortex stretching due to cross-isobath flow and
is thus a measure of the net upwelling in the region.

## 5    Model Prediction of Mean Circulation and Validation Using Geodetically Estimated MDT

Before the dynamical role of $\Delta\eta_c$ is explored in the realistic, high-resolution regional ocean model GoMSS, its predictions of
the MDT and mean circulation are presented. To illustrate the main features of the circulation in the Scotian Shelf and Gulf of
Maine region, the mean depth-averaged currents predicted by GoMSS for the period 2011–2013 are shown in Figure 5.

GoMSS is able to capture the main features of the mean circulation which are closely connected to the complex bathymetry
in the region and have been documented in numerous studies. The nearshore outflow from the Gulf of Saint Lawrence through





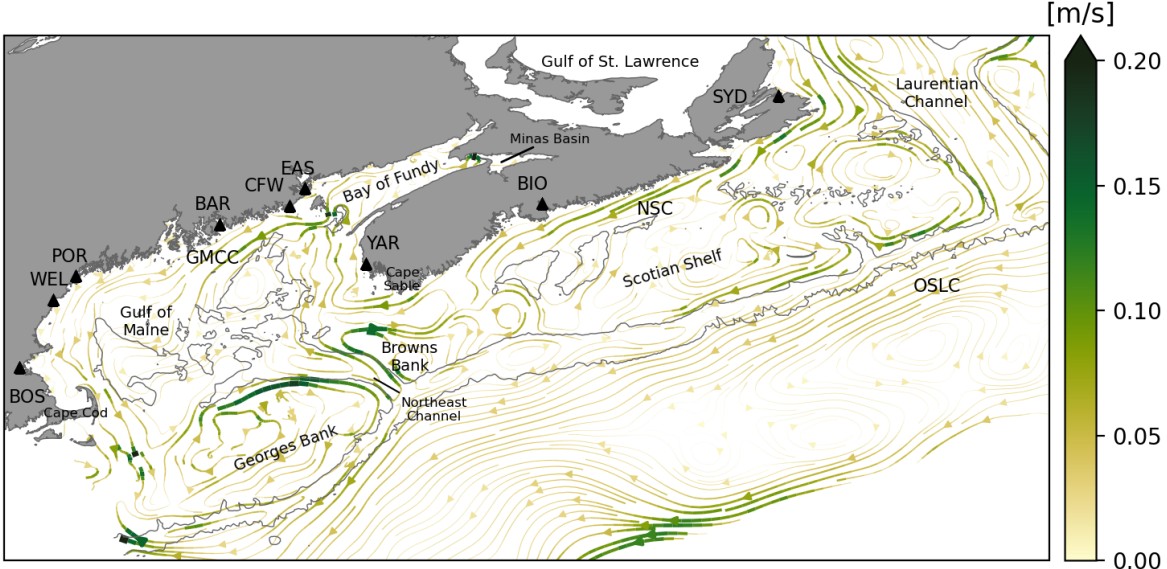

**Figure 5.** Streamlines of mean depth-averaged circulation predicted by GoMSS for the period 2011–2013. Grey contours mark the 200 and 2000 m isobaths and triangles show the locations of the tide gauges listed in Table 1. Acronyms indicate circulation features described in the text. NSC: Nova Scotia Current; OSLC: Offshore Labrador Current; GMCC: Gulf of Maine Coastal Current.

Cabot Strait is the origin of the Nova Scotia Current (NSC) which follows the coastline toward the Gulf of Maine. This outflow is associated with relatively fresh and cold water originating from the inshore Labrador Current and runoff from the Saint Lawrence River (e.g., Smith and Schwing, 1991; Hannah et al., 2001; Dever et al., 2016; Rutherford and Fennel, 2018).

Another part of the outflow through Cabot Strait follows the western side of the Laurentian Channel and joins the offshore branch of the Labrador Current (OSLC) flowing along the shelf break. The strong current along the offshore boundary of GoMSS is related to mesoscale eddies associated with the Gulf Stream outside the model domain and is also present in the forcing data from GLORYS12v1 (not shown).

On the shelf and in the Gulf of Maine, the mean circulation is dominated by rectified tidal flow which is aligned with
bathymetric features. Most notable is the clockwise gyre on Georges Bank with predicted residual currents up to $20\,\mathrm{cm\,s^{-1}}$ along its northern flank. This is consistent with previous studies and can be explained by tidal rectification and baroclinic processes associated with strong tidal mixing (Loder, 1980; Butman et al., 1982; Greenberg, 1983; Naimie et al., 1994; Naimie, 1996). GoMSS also predicts a clockwise gyre over Browns Bank which is caused by the same mechanisms (e.g., Greenberg, 1983; Smith, 1983; Tee et al., 1993; Hannah et al., 2001). These two gyres create an inflow-outflow pattern in the Northeast
Channel.





In the vicinity of Cape Sable, strong tidal currents generate a tidally rectified mean flow that locally enhances the Nova Scotia Current. It has been shown that this is also associated with permanent topographic upwelling in that region (Garrett and Loucks, 1976; Greenberg, 1983; Tee et al., 1988, 1993; Chegini et al., 2018).

In the Gulf of Maine, GoMSS predicts a generally counter-clockwise circulation. One dominant feature is the Gulf of Maine Coastal Current (GMCC) flowing from the Bay of Fundy along the coast of Maine and splitting into two branches south of Bar Harbor (BAR). This pattern is consistent with observations and is primarily driven by a pressure gradient force (Pettigrew et al., 1998, 2005).

The circulation features described above are also expressed in the MDT predicted by GoMSS (Figure 6a). To center MDT variations on the shelf (water depths $< 200\,\mathrm{m}$) around zero, the spatial median value over this region has been subtracted. Contours indicate the 200 and 2000 m isobaths which mark the shelf break as well as important banks and channels on the shelf. Triangles show the locations of the tide gauges listed in Table 1.

The strong signal in the deep ocean is related to eddies associated with the Gulf Stream and the offshore branch of the Labrador Current. On the shelves, gradients in MDT are generally aligned with bathymetric features which is consistent with the topographically driven and tidally rectified circulation described above.

Relatively high values of MDT are predicted on the western side of Cabot Strait associated with the outflow from the Gulf of Saint Lawrence. The offshore gradient of MDT indicates a geostrophic balance with the Nova Scotia Current. Additionally, areas of elevated MDT are apparent over the banks on the shelf driven by tidal rectification. In the Gulf of Maine, MDT is generally lower toward the center which is consistent with the overall counterclockwise circulation. This is also in agreement with observations (Li et al., 2014a). As shown by Renkl and Thompson (2022), the predicted MDT in the upper Bay of Fundy has to be treated with caution because of the limited spatial resolution of GoMSS in that region.

## 5.1 Model Validation Using Geodetic Tilt Estimates

In Figure 6b, the predicted and observed MDT along the coast are shown as a function of alongshore distance from Cape Cod to Cabot Strait. The means of the observations and predictions of coastal MDT at the grid points closest to the tide gauges have been removed.

The shaded area marks the coastline in the upper Bay of Fundy and illustrates more clearly the strong setdown in that area. As discussed above, the MDT prediction in that region has to be treated with caution and therefore, the coastline is separated in two parts along the Gulf of Maine (Figure 6c) and Nova Scotia (Figure 6d), respectively. In both panels, the means of the respective observations and predictions at the grid points closest to the tide gauges have been removed.

Along the coast of the Gulf of Maine, the predicted coastal MDT is mostly flat, with a small increase toward Cape Cod Bay due to wind setup. The small-scale variability originates from local interactions between the flow and bathymetry in tidal inlets which are part of the rugged coastline. While the predicted local minimum near Cutler Farris Wharf (CFW) and Eastport (EAS) is due to local processes, the overall difference in MDT either side of this setdown is associated with the Gulf of Maine Coastal Current.



**Figure 6.** Predicted and observed mean dynamic topography (MDT). (a) MDT predicted by GoMSS (spatial median value over area where water depth $< 200\,\mathrm{m}$ removed). Markers indicate the locations of the coastal tide gauges listed in Table 1. The line separates the upper Bay of Fundy where the model has difficulty resolving the residual circulation due to the limited resolution. (b) Coastal MDT as a function of distance along the coast of Gulf of Maine and Nova Scotia. The minima in Minas Passage ($-28\,\mathrm{cm}$ and $25\,\mathrm{cm}$, respectively) are not shown. Geodetic estimates of MDT are shown with their respective uncertainty. The shaded area indicates the coast along the upper Bay of Fundy. (c) and (d) Enlarged views of either side of the shaded area in (b). In panels (b)–(d), the means of the respective observations and predictions at the grid points closest to the tide gauges have been removed.





The alongshore MDT predicted along the coast of the Gulf of Maine agrees well with the geodetic estimates. The largest
discrepancy is found at Boston (BOS) where the tide gauge is located inside the harbor, sheltered from the open ocean.
Therefore, it is likely that the strong setup seen in the observations is a manifestation of local processes. However, it cannot be
ruled out that the wind setup toward Cape Cod Bay is underestimated in GoMSS.

Along the coast of Nova Scotia (Figure 6d), both the observations and predictions show a strong tilt of coastal MDT. The
observed difference in MDT between the tide gauges in Sydney (SYD) and Yarmouth (YAR) is $\Delta\eta_c = 5.9 \pm 2.0\,\mathrm{cm}$. GoMSS
predicts a tilt of $\Delta\eta_c = 8.3\,\mathrm{cm}$. This is slightly larger than the geodetically estimated tilt, but within two standard deviations
of the observed value. Note that a local setdown near YAR is predicted which is related to strong tidal currents in that region.
The tide gauge itself is located inside Yarmouth Harbour which is not resolved in the model. This will lead to discrepancies
between the model and observations.

Rather than stating $\Delta\eta_c$ as a difference between two fixed locations, it is often reported as the alongshore gradient consistent
with its expression in the momentum equation. However, it is not straightforward to calculate the coastal MDT gradient because
the irregular shape of the coastline leads to uncertainty in the distance between the fixed locations (Mandelbrot, 1982). For
example, using an alongshore distance of $\Delta L = 999\,\mathrm{km}$ computed from the coastline in GoMSS results in a predicted MDT
gradient $\Delta\eta_c/\Delta L = 8.4 \times 10^{-8}$ (equivalent to $0.8\,\mathrm{cm}$ per $100\,\mathrm{km}$). If the interest is the large-scale gradient, this value is
arguably an underestimation. If instead one were to use $\Delta L = 650\,\mathrm{km}$ based on three straight line segments from SYD to
YAR, the gradient is $\Delta\eta_c/\Delta L = 1.3 \times 10^{-7}$. This gradient is comparable to the values used by Smith (1983) in his diagnostic
model to describe the circulation off southwest Nova Scotia. However, the above discussion highlights the subjectivity that can
be introduced by focusing on gradients rather than $\Delta\eta_c$ between two fixed locations.

In addition to the large-scale tilt, GoMSS predicts local minima of MDT around YAR and just southeast of it at Cape Sable.
As discussed above in relation to the mean circulation, these setdowns can be explained by the strong tidal currents and the
curvature of the coastline in that region (e.g., Greenberg, 1983; Smith, 1983; Tee et al., 1993; Chegini et al., 2018).

The above discussion answers the first major question raised in the Introduction: Can new observations of geodetically
referenced coastal sea level help validate high-resolution regional ocean models? The good agreement of $\Delta\eta_c$ estimated in-
dependently by the hydrodynamic and geodetic approaches provides validation of the ocean model. The agreement gives
confidence that GoMSS captures the mean circulation, including the effect of tidal rectification, on the Scotian Shelf and in the
Gulf of Maine. In the next two sections, the following questions will be addressed: What can the alongshore tilt of MDT at the
coast tell us about shelf circulation? What are the implications for coastal monitoring?

## 6  Predicted Mean Alongshore Momentum Balance

As discussed in Section 3.1, the large-scale tilt of alongshore MDT at the coast is expected to be balanced by the sum of wind
stress, bottom friction, and lateral mixing. Using output from GoMSS, it is possible to check if this balance holds in the model
and identify the dominant processes that lead to the predicted alongshore tilt of MDT. This is a necessary step before using
$\Delta\eta_c$ to make inferences about coastal and regional circulation.



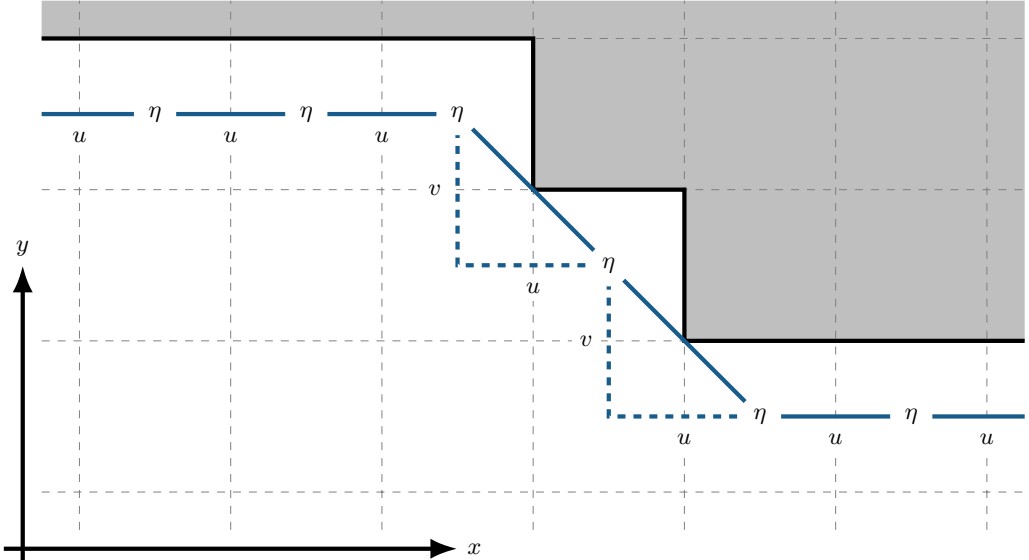

**Figure 7.** Schematic of the alongshore integration path in GoMSS. The gray area marks the land and the solid black line illustrates the coastline in the model. Solid blue lines illustrate segments of the integration path between grid cells and dashed lines indicate components of diagonal segments.

Here, the approach of Lin et al. (2015) is adopted, where each term in the alongshore momentum equation is integrated separately along the coast to yield an equivalent change in sea level. This approach is preferable over the comparison of the actual terms in the momentum equation which can be noisy due to local variations in bathymetry and coastline. Alongshore
integration smooths out these small-scale fluctuations and makes results easier to interpret.

The alongshore integration path $C_c$ is defined such that it connects all center points of the grid cells closest to the coast where MDT is defined in the model (Figure 7). Note that the coastline in the model follows the edges of the grid cells and therefore, $C_c$ is half a grid cell away from the coast. Due to the grid structure, the coastline in the model has a step-like shape, however, the integration path runs diagonally as indicated in the schematic.

Due to the staggering of the variables on the Arakawa C-grid, the alongshore integral of the momentum equation is straight-forward. The approximation of the line integral $\int \mathbf{u}(x, y) \cdot d\mathbf{r}$ is illustrated by the schematic in Figure 7. The $x$- and $y$-components of the momentum equation are defined at the $u$- and $v$-points, respectively, on the model grid. Each component is multiplied by the appropriate grid spacing $\Delta x$ or $\Delta y$ and summed up along the integration path. For increments in $x$-direction, $\Delta y = 0$ and for steps in $y$-direction, $\Delta x = 0$. Diagonal elements include both the $u$- and $v$-component as shown in Figure 7.

Prior to the alongshore integration, the output fields of the three-dimensional momentum trends were first depth-averaged and then averaged over the period 2011–2013.

Figure 8 shows the alongshore MDT as well as mean sea level contributions by the individual terms in the momentum equation at the coast on the Scotian Shelf. Note that the mean of each term over the shown segment was subtracted to center



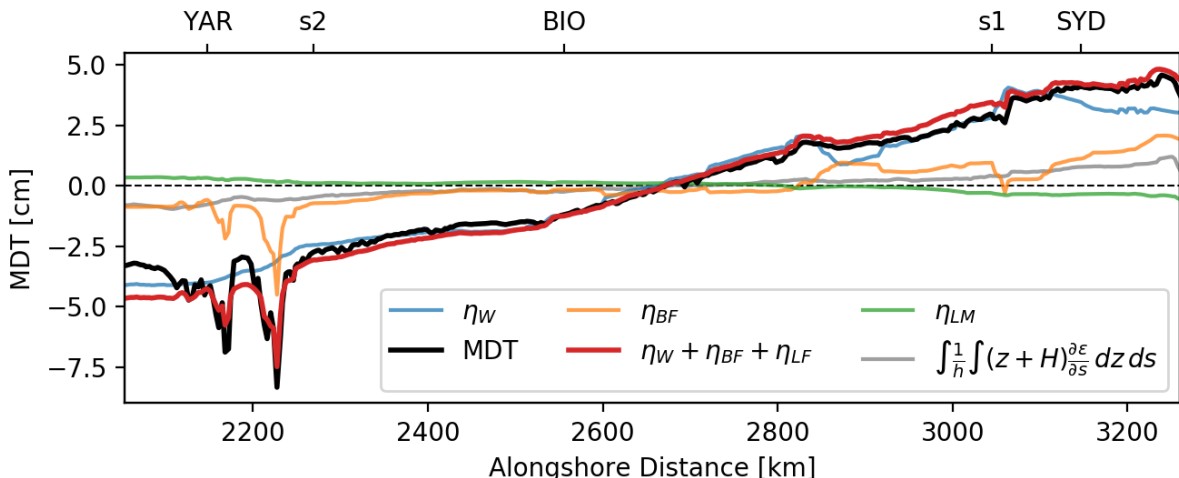

**Figure 8.** Predicted MDT (black line) and contributions by individual terms in the alongshore, depth-averaged momentum balance at the coast on the Scotian Shelf: wind stress ($\eta_\mathrm{W}$), bottom friction ($\eta_\mathrm{BF}$), and lateral mixing ($\eta_\mathrm{LM}$) as well as their sum. The contribution from the depth-averaged baroclinic pressure gradient is also shown (grey line). Note that the mean of each term has been subtracted to center the curves around zero. Alongshore locations of the tide gauges are shown by their respective abbreviation.

the curves around zero. The red line shows the sum of the contributions of wind stress, bottom friction, and lateral mixing. It
is clearly in close agreement with the MDT along the coast predicted by GoMSS. Including the remaining terms effectively
closes the momentum balance defined by (3).

The alongshore wind stress causes the large-scale setup of MDT ($\eta_\mathrm{W}$) with higher values toward Cabot Strait. The wind setup
is partially balanced by bottom friction ($\eta_\mathrm{BF}$) acting on the current at the coast. In the region around Sydney, bottom drag is
strongest and associated with the Nova Scotia Current flowing close to the coast as it exits the Gulf of Saint Lawrence. Further
downstream, the Nova Scotia Current veers offshore and bottom friction at the coast becomes negligible except in the region
around YAR where it balances local MDT minima. These features are due to the Bernoulli effect caused by strong tidal currents
around Cape Sable mentioned above. Variations of the sea level equivalent due to lateral mixing ($\eta_\mathrm{LM}$) are relatively small along
the coast of Nova Scotia. These results show that $\Delta\eta_\mathrm{c}$ is primarily a response to local Ekman dynamics and spatial variations of
bathymetry (Lentz and Fewings, 2012). It is important to note that the bottom friction term depends on the alongshore current
and thus implicitly includes the effect of large-scale, non-local forcing (e.g., JEBAR).

The close agreement between the coastal MDT and the sum of sea level equivalents due to the wind stress, bottom friction,
and lateral mixing predicted by GoMSS is consistent with observational studies for other regions along the eastern seaboard of
North America (e.g., Scott and Csanady, 1976; Fewings and Lentz, 2010). They show that the coastal circulation is generally in
"frictional equilibrium" (Lentz and Fewings, 2012). The overriding importance of wind stress and bottom friction also justifies
the choice of Csanady's arrested topographic wave model in Section 4.





Previous studies have shown that the wind forcing is the dominant driver of sea level variability on the Scotian Shelf on synoptic to interannual timescales (e.g., Thompson, 1986; Schwing, 1989; Li et al., 2014b). Many of these studies also demonstrate the influence of remote forcing and coastally trapped waves propagating along the Scotian Shelf. This is evident in Figure 8 for alongshore distances $> 3100\,\mathrm{km}$ which corresponds to the coastline between SYD and the open boundary across Cabot Strait.

Here, the wind-driven sea level tilts in the opposite direction compared to the MDT which is primarily balanced by bottom friction.

The steric contribution to the alongshore tilt of MDT at the coast is small ($0.9\,\mathrm{cm}$ between $s_1$ and $s_2$, about half of the contribution by bottom friction). This justifies the assumption made in the simplified alongshore momentum equation (8). In the cross-shore direction, a large density gradient exists which is related to the geostrophic outflow from the Gulf of Saint

Lawrence with a coastal setup of MDT on the western side of Cabot Strait (El-Sabh, 1977). According to the idealized model of Csanady (1982, Figure 4), this setup "diffuses" in the downstream direction, with the flow trapped within a widening coastal boundary layer. Note that the associated fanning out of MDT contours is evident in Figure 6a which can be compared with the region $y < 0$ in Figure 4.

## 7  Coastal and Regional Interpretations of $\Delta\eta_{\mathrm{c}}$

In Section 3 it was shown that the alongshore tilt of MDT at the coast can be interpreted in terms of the coastal and the regional circulation. Using idealized models, it was demonstrated that $\Delta\eta_{\mathrm{c}}$ is a measure of the mean alongshore current at the coast (coastal view), but can also be related to the net upwelling due to vortex stretching offshore (regional view). Here, it will be tested whether these views hold in GoMSS with a focus on the nearshore region between the reference points $s_1$ and $s_2$ outlined by the red polygon in Figure 1. Note that the coastal and regional views are based on time-averaged dynamics and can

therefore be applied to shelf circulation on timescales where a quasi-steady state can be assumed.

### 7.1  Coastal View

Based on (9), $\Delta\eta_{\mathrm{c}}$ can be related to the integrated frictional effects along the coast. As shown above, alongshore wind stress is the main contributor to the MDT difference at the coast of Nova Scotia. Since the sea level equivalent due to wind stress can be computed from the GoMSS model output, the special case in (11) will be used. Given the negligible role of lateral mixing

and assuming linear bottom friction $\tau_s^{\mathrm{b}} = \lambda \overline{u}_s$, the wind-corrected tilt of MDT is proportional to the mean depth-averaged alongshore current

$$\langle u_s \rangle = \frac{1}{\Delta L} \int\limits_{s_1}^{s_2} \overline{u}_s \, ds. \tag{40}$$

It follows from (11) that the predicted mean depth-averaged alongshore current based on the $\Delta\tilde{\eta}_{\mathrm{c}}$ is

$$\langle \tilde{u}_s \rangle = -\frac{g H_0}{\lambda \Delta L} \Delta\tilde{\eta}_{\mathrm{c}}, \tag{41}$$

where $H_0$ is the mean depth of the model along the coast.





It is to be expected that $\lambda$ changes with seasonal stratification of the water column and therefore a time-varying friction coefficient is defined by

$$\lambda = \lambda_0\{1 + \alpha\cos[(t-t_0)\omega]\}, \tag{42}$$

where $\lambda_0$ is a constant drag coefficient, $\alpha$ is a factor that controls the amplitude of the seasonal variations with frequency
$\omega = 2\pi/365$ days$^{-1}$, and $t_0$ corresponds to a time when stratification is at its seasonal maximum.

Furthermore, defining $\kappa = gH_0/\lambda_0\Delta L$, (41) becomes

$$\langle \tilde{u}_s \rangle = -\frac{\kappa}{1 + \alpha\cos[(t-t_0)\omega]}\Delta\tilde{\eta}_{\text{c}}, \tag{43}$$

which is a model with three parameters that can be applied to estimate the mean alongshore current based on $\Delta\tilde{\eta}_{\text{c}}$.

Figure 9a shows time series of $\langle u_s \rangle$ and $\langle \tilde{u}_s \rangle$ based on daily mean model output from GoMSS with realistic values of
$\lambda_0 = 1.3 \times 10^{-3}\,\mathrm{m\,s}^{-1}$, $\alpha = 0.5$, and $t_0 = 30$ days. These values were chosen to yield maximum agreement between the time series and $\lambda_0$ is comparable to literature values (e.g., Csanady, 1982). The mean water depth along the coast, $H_0 = 23.4\,\mathrm{m}$, and $\Delta L = 774\,\mathrm{km}$ were directly computed from the GoMSS grid. Based on the definition in (40), positive values correspond to a southwest flow from $s_1$ to $s_2$. Periodograms were analyzed to check if the time series contain an aliased signal from tidal variations. It was found that there is no significant energy at the alias frequencies. A third order Butterworth lowpass
filter with cutoff frequency of 15 days was applied to the time series to remove high-frequency variability and thereby allow a quasi-steady state to be assumed.

Both time series show coherent low-frequency variability for timescales of 15 days or longer with correlation $r = 0.92$. The RMSE between the time series is $1.4\,\mathrm{cm\,s}^{-1}$. The good agreement between the two time series indicates that mean strength of the alongshore current can be estimated by the MDT difference at the coast after correction for the local wind effect. However,
there is a small offset between the two time series that can be explained by the alongshore gradient of depth-integrated potential energy anomaly. While this term typically vanishes in shallow water, it is nonzero in GoMSS due to the finite water depth along the coast in the model (not shown).

### 7.2 Regional View

Equation (22) relates coastal MDT to area integrated upwelling as well as wind stress and frictional forces projected along
the offshore boundary. Assuming the wind stress and frictional terms are negligible in deep water because of their inverse dependence on $H$, the tilt of sea level along the coast is then given by

$$g\Delta\eta_{\text{c}} = f\iint\limits_{A} \frac{w^*}{H}\,dA. \tag{44}$$

To physically interpret this equation, note $H/w^*$ is the time it takes for a water parcel moving vertically at $w^*$ to travel from the seafloor to the surface. This motivates the following definition of an area mean "upwelling" rate:

$$\xi = \frac{1}{A}\iint\limits_{A} \frac{w^*}{H}\,dA. \tag{45}$$





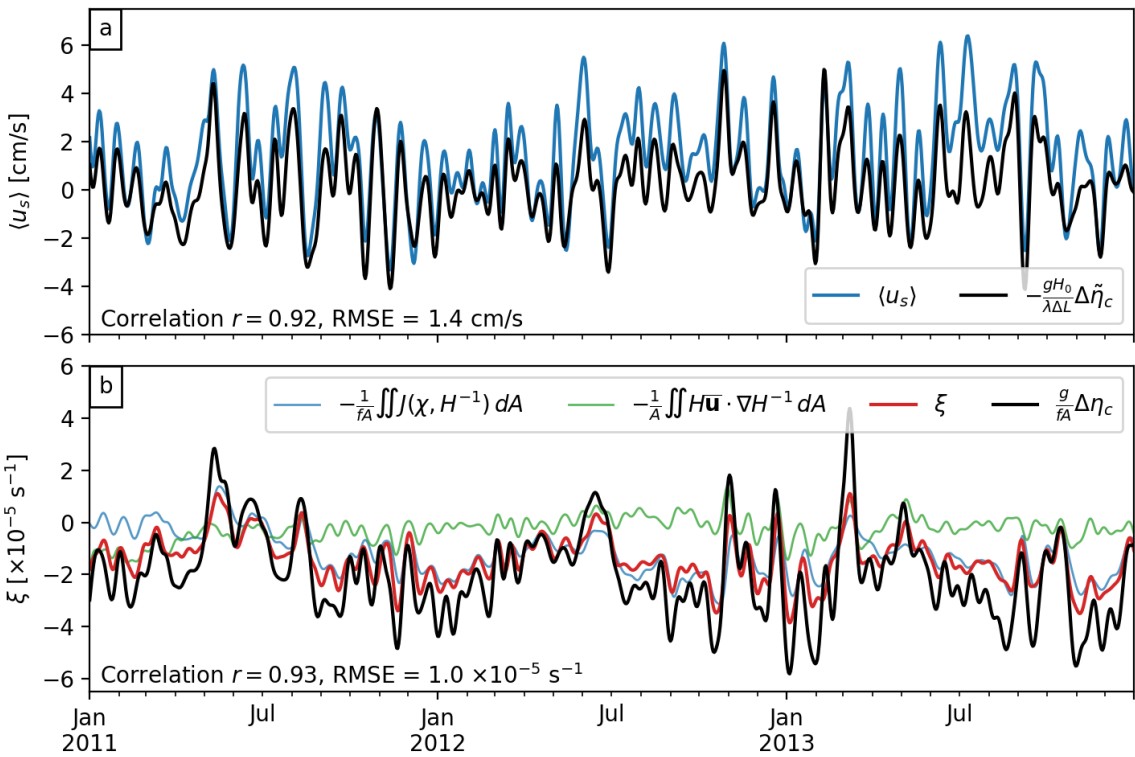

**Figure 9.** Lowpass-filtered time series of the alongshore tilt of MDT at the coast of Nova Scotia and related quantities. a) Mean alongshore depth-averaged current predicted by GoMSS (blue) and estimated from wind-corrected tilt of predicted MDT (black). b) Area-integrated JEBAR (blue) and flow across isobaths (green) as well as resulting upwelling rate (red). The estimated upwelling rate based on $\Delta\tilde{\eta}_c$ is shown in black.

Substituting this definition of $\xi$ into (44) gives the following expression for the upwelling rate in terms of the tilt of MDT along the coast:

$$\xi = \frac{g}{fA}\Delta\eta_c. \tag{46}$$

This equation can be used to estimate the upwelling rate directly from the alongshore tilt of MDT at the coast.

Equation (45) has been used to estimate time series of $\xi$ from daily mean output from GoMSS integrated over the area enclosed by the red polygon in Figure 1. High-frequency variability was removed using the same lowpass filter described above. The resulting time series is shown by the red line in Figure 9b. $\xi$ was also estimated from $\Delta\eta_c$ using (46) with and $A = 27536.7\,\mathrm{km}^2$ and $f = 10^{-4}\,\mathrm{s}^{-1}$. This estimate will henceforth be denoted by $\tilde{\xi}$.

There is clearly close agreement between $\xi$ and $\tilde{\xi}$ in terms of both correlation ($r = 0.93$) and RMSE ($1.0 \times 10^{-5}\,\mathrm{s}^{-1}$). The
estimated $\tilde{\xi}$ is generally larger than $\xi$ during extreme events. This can be explained by the assumptions underlying (46), i.e., the neglect of the wind stress and frictional terms in deep water. Additional analysis (not shown) indicates that the wind setup





along the offshore boundaries is not necessarily zero and explains most of the differences between $\xi$ and $\tilde{\xi}$. However, the good agreement of the time series demonstrates that $\Delta\eta_c$ is an effective measure of the net regional upwelling in a realistic model.

Figure 9b also shows the two components of $\xi$ defined by (19): the area-averaged JEBAR (blue) and the vortex stretching due to depth-averaged flow across isobaths (green). Clearly, the JEBAR contribution is dominant thereby highlighting the importance of baroclinicity in driving the Nova Scotia Current. The relatively fresh outflow from the Gulf of Saint Lawrence through Cabot Strait causes a strong cross-shore density gradient leading to a geostrophic flow along the coast.

In general, $\xi$ is negative which is consistent with an overall vortex squashing by an onshore flow across isobaths. Episodes where $\xi > 0$ can be identified which correspond to periods of offshore flow and associated net downwelling.

As shown above, the mean alongshore wind stress leads to a MDT setup along the coast, but it also causes an offshore Ekman transport in the surface boundary layer. Below, a mean onshore flow leads to upwelling of subsurface water at the coast. Although the wind stress can be uniform over a large area, the increase in water depth offshore leads to an input of negative vorticity into the water column. The cross-isobath flow towards shallower water depth ensures that potential vorticity is conserved.

Similarly, frictional forces at the bottom as well as JEBAR exert a torque on the water column and generate relative vorticity. While bottom friction leads to an offshore flow across isobaths (see idealized case in Section 4.2.2), JEBAR is dominant along the coast of Nova Scotia. Here it drives an onshore flow which is captured in the time series of the upwelling rate $\xi$.

## 8 Summary and Conclusion

In this study, we used newly available geodetic estimates of coastal MDT to validate the GoMSS regional ocean model. Additionally, the relationship between coastal MDT and shelf circulation was studied using a combination of theory, idealized models, and a numerical ocean circulation model (GoMSS). It was first shown that GoMSS predicts the main features of the mean circulation that are known to exist on the Scotian Shelf and in the Gulf of Maine, including the effect of outflow from the Gulf of Saint Lawrence and tidal rectification. While the coastal MDT is generally flat in the Gulf of Maine, GoMSS predicts a MDT difference between North Sydney and Yarmouth of $\Delta\eta_c = 8.3\,\mathrm{cm}$. This is slightly larger than the geodetically determined value of $\Delta\eta_c = 5.9 \pm 2.0\,\mathrm{cm}$, but the difference is not statistically significant.

These results lead to an affirmative answer to the first question raised in the Introduction: *Can new observations of geodetically referenced coastal sea level help validate high-resolution regional ocean models like GoMSS?* The good agreement of the independent estimates of MDT derived from the hydrodynamic and geodetic approaches provides validation of both the ocean and geoid models used in this study. However, the use of $\Delta\eta_c$ for model validation is limited to regions with long, geodetically referenced sea level records. In the upper Bay of Fundy, GoMSS predicts an unusually strong setdown in MDT that could not be directly validated because no sufficiently long sea level records exist (see Renkl and Thompson (2022) for a detailed study of this region).

The other questions addressed in this chapter focus on the physical interpretation of $\Delta\eta_c$: *What can the alongshore tilt of MDT at the coast tell us about shelf circulation? What are the implications for coastal monitoring?* Based on theory





and idealized models of ocean and shelf circulation, it was shown that $\Delta\eta_c$ can be interpreted in two complementary, and dynamically consistent, ways. The coastal view is based on the time-averaged alongshore momentum equation at the coast and the regional view is based on vorticity dynamics integrated over an adjacent offshore region. The idealized ocean model of Stommel (1948) was used to show that $\Delta\eta_c$ can be used to estimate the coastal flow averaged along the western boundary, and also the basin-averaged, wind-forced upwelling (and hence the integrated Sverdrup transport). Furthermore, the coastally

trapped wave model of Csanady (1982) was used to show that $\Delta\eta_c$ can be used to estimate spatially averaged upwelling caused by depth-averaged flow across a linearly sloping bathymetry.

The usefulness of the coastal and regional views was demonstrated in a more realistic setting using output from GoMSS. First, it was shown that the tilt of MDT along the coast of Nova Scotia is balanced primarily by wind stress, bottom friction, and a relatively small contribution from lateral mixing. This "frictional equilibrium" is a general characteristic of coastal circulation

(Lentz and Fewings, 2012). This simplified momentum balance means that, if the wind setup is known, $\Delta\eta_c$ can provide a direct estimate of the average alongshore current $\langle\hat{u}_s\rangle$ between two points at the coast under the assumption that bottom friction can be approximated by a linear dependence on the depth-averaged flow. The scale factor linking $\Delta\eta_c$ and $\langle\hat{u}_s\rangle$ depends only on the mean water depth at the coast and the linear bottom drag coefficient.

The regional view is more subtle than the coastal view. As demonstrated with the idealized models, $\Delta\eta_c$ can be used to

approximate the upwelling averaged over an offshore area. For the regional view to be physically meaningful, this area needs to be limited to a flow regime in the nearshore such that the dominant momentum balance simplifies along the offshore integration path. In the illustration of the realistic case using GoMSS, we chose the outer edge of the integration path in a quiescent region outside the Nova Scotia Current where the water is deep enough that wind stress and frictional terms can be neglected. Where the current crosses the integration path, it is in geostrophic balance.

On the Scotian Shelf, the upwelling rate (denoted by $\xi$) is related to vortex tube stretching caused by the combined effect of JEBAR and depth-averaged flow across isobaths. JEBAR plays a critical role and causes an overall onshore transport near the sea floor which can result in upwelling at the coast of Nova Scotia. However, there are brief periods of wind-induced downwelling.

The relationship between $\Delta\eta_c$ and the coastal and regional circulation applies not only to the long-term mean, but also on

timescales for which a quasi-steady state can be assumed. Time series of $\langle u_s\rangle$ and $\xi$ were calculated directly from GoMSS output and also estimated from $\Delta\eta_c$ using two simple linear relationships resulting from the coastal and regional views. The parameters in these linear relationships are based on physics. The tilt-based estimates are in good agreement with the filtered time series of $\langle u_s\rangle$ and $\xi$ calculated directly from model output.

This has obvious implications for ocean monitoring using geodetically referenced sea level observations recorded by coastal

tide gauges. For example it may be possible to use long coastal sea level records to estimate time series of upwelling rates. Such information may be of interest to biological oceanographers interested in understanding changes in nutrient cycling on the shelf over recent decades. This speculation applies not only to the Scotian Shelf. For example, in the future, it would be interesting to test the idea on the west coast of North America given the large number of long, geodetically referenced sea level records (e.g., Lin et al., 2015) and the large amount of hydrographic data, e.g., the California Cooperative Oceanic Fisheries (CalCOFi)

Database. Finally, our findings have implications for future deployments of tide gauges if they are to be used to monitor the shelf-scale ocean circulation. Coastal MDT can be affected by local processes, e.g., strong tidal flow around headlands and therefore, the location of the tide gauges should be exposed to the open ocean and at distance to areas where local processes dominate.

*Code and data availability.* Sea level observations, benchmark sheets, and GPS ellipsoidal heights for tide gauges in the USA were ob-
tained from NOAA (https://tidesandcurrents.noaa.gov/) and the Online Positioning User Service (OPUS) provided by the NGS (https://geodesy.noaa.gov/OPUS/view.jsp). The conversion of the vertical datum was performed using the Horizontal Time-Dependent Positioning tool (HTDP; https://geodesy.noaa.gov/TOOLS/Htdp/Htdp.shtml). For tide gauges in Canada, sea level observations and benchmark sheets were obtained from the CHS (https://www.isdm-gdsi.gc.ca/isdm-gdsi/twl-mne/maps-cartes/inventory-inventaire-eng.asp, https://tides.gc.ca/en/stations) and GPS ellipsoidal heights were retrieved from NRCAN (https://webapp.csrs-scrs.nrcan-rncan.gc.ca/geod/data-donnees/
passive-passif.php). The Canadian Gravimetric Geoid model of 2013 (CGG2013a) can be retrieved from https://webapp.csrs-scrs.nrcan-rncan.gc.ca/geod/data-donnees/geoid.php. All of the code and data required to configure and run the GoMSS model are publicly available: NEMO source code (https://www.nemo-ocean.eu/), NCEP CFSv2 data for surface boundary forcing (https://rda.ucar.edu/datasets/ds094.0/), GLORYS12v1 for open boundary conditions (https://data.marine.copernicus.eu/product/GLOBAL_MULTIYEAR_PHY_001_030), and FES2004 for tidal boundary forcing (https://www.aviso.altimetry.fr/en/data/products/auxiliary-products/global-tide-fes.html).

*Author contributions.* CR and KRT conceptualized the study; CR compiled the observations, performed model simulations, analyzed the data, and wrote the manuscript draft with guidance from KRT and ECJO; CR and ECJO reviewed and edited the manuscript.

*Competing interests.* The authors declare that they have no conflict of interest.

*Acknowledgements.* This work was funded by the Marine Environmental Observation, Prediction and Response (MEOPAR) Network of Canada. CR acknowledges funding from Nova Scotia Graduate Research Scholarship. The authors thank Anna Katavouta and Simon Hig-
ginson for their help with the model configuration and data acquisition.



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



**Table 1.** Summary of geodetic MDT observations in the study area for the period 2011–2013. The numbers in parentheses after the station name are the IDs in the NOAA and CHS databases, respectively. The columns Lat. and Lon. list coordinates of the tide gauges. BM ID refers to the permanent identifiers of the GPS benchmarks assigned by NGS and NRCAN and the ellipsoidal height at these benchmarks are listed under $h_e$. The geoid height $N$ interpolated to the benchmark locations are listed in column CGG2013a. In the last column, the resulting geodetic estimates of MDT are presented. Their error is a combination of the errors in $h_e$ and $N$ and were calculated using standard error propagation rules.

| | Name | Lat. [°N] | Lon. [°E] | BM ID | $h_e$ [m] | CGG2013a [m] | MDT [m] |
|---|---|---|---|---|---|---|---|
| SYD | North Sydney (612) | 46.2167 | -60.2500 | NS29101 | 29.502840 ± 0.010 | -13.180 ± 0.012 | -0.201 ± 0.016 |
| BIO | Bedford Institute (491) | 44.6833 | -63.6167 | 961000 | 3.099574 ± 0.000 | -21.469 ± 0.012 | -0.224 ± 0.012 |
| YAR | Yarmouth (365) | 43.8333 | -66.1167 | XXN9007 | -19.457698 ± 0.015 | -23.201 ± 0.011 | -0.260 ± 0.019 |
| EAS | Eastport (8410140) | 44.9033 | -66.9850 | PD1179 | -18.828134 ± 0.006 | -23.303 ± 0.013 | -0.321 ± 0.015 |
| CFW | Cutler Farris Whf (8411060) | 44.6570 | -67.2047 | PD0497 | -16.757344 ± 0.014 | -23.243 ± 0.022 | -0.309 ± 0.026 |
| BAR | Bar Harbor (8413320) | 44.3917 | -68.2050 | BBGN12 | -18.875490 ± 0.004 | -24.797 ± 0.016 | -0.305 ± 0.017 |
| POR | Portland (8418150) | 43.6567 | -70.2467 | AJ2726 | -24.168144 ± 0.004 | -26.996 ± 0.012 | -0.298 ± 0.012 |
| WEL | Wells (8419317) | 43.3200 | -70.5633 | BBCF81 | -23.691065 ± 0.007 | -27.333 ± 0.013 | -0.312 ± 0.015 |
| BOS | Boston (8443970) | 42.3539 | -71.0503 | AJ4030 | -26.394000 ± 0.005 | -28.666 ± 0.010 | -0.266 ± 0.011 |