# Peer review of "The Alongshore Tilt of Mean Dynamic Topography and its Implications for Model Validation and Ocean Monitoring"

_EGUsphere, 2024_

## Author Comment (AC2)

**The Alongshore Tilt of Mean Dynamic Topography and its Implications for Model Validation and Ocean Monitoring**

Christoph Renkl, Eric C.J. Oliver and Keith R. Thompson

**Response to Comments by Reviewers**

We thank both reviewers for their insightful comments. We have responded to all the comments as detailed below. The labels to the comments of Reviewer 1 are of the form [R1.x.y], where y is the number of the comment in Section x of the review. Similarly, labels to the comments of Reviewer 2 are of the form [R2.x].The reviewers' comments are repeated below in bold font, our responses are given regular font with, and modified parts of the text including the respective line numbers is indicated by italicized font. We believe the changes have resulted in a significant improvement to the manuscript and hope the reviewers and editor will find the revised version suitable for publication.

**Reviewer 1**

**This paper covers quite a lot of ground. It presents a model calculation of the mean dynamic topography along the Scotian Shelf - Gulf of Maine region, including comparison with tide gauge observations and interpretation in terms of terms in the equation of motion along the coastline (which in the model is at a mean depth of 23.4 m). These sections of the paper are a nice piece of work in themselves, similar in spirit to the work of Lin et al. (2015) as cited by the authors.**

**However, the scope is much broader, including a theoretical section relating the alongshore sea level slopes to offshore processes, sections on the Stommel model and on Csanady's Arrested Topographic Wave, and interpretation in terms of a regional average of upwelling, followed by a test of this interpretation using diagnostics of the time-varying component of the ocean model. I find this broader aspect of the paper unconvincing, and the presentation rather disorganised. The maths appears to be correct (except that the wind stress terms should all be divided by $\rho_0$ and the quickly-abandoned nonlinear terms are incorrect), but the interpretation and link with other models is not clear and, in particular, the description of the relevant diagnostic as an area-averaged upwelling does not seem appropriate. Accordingly, my recommendation for the paper is that it needs a major revision.**

Thank you very much for your constructive comments and feedback on our paper! Following your suggestions, we reorganized the manuscript, simplified the maths, and clarified the interpretations and links between the considered models as detailed below.

**R1.1 Interpretation and link to idealised models**

**[R1.1.1] The crucial diagnostic derived in section 3 is $-(u - u_g + u_{gb})\cdot\nabla H$, which is described as an upwelling, presumably on the basis that the terms other than $u_g\cdot\nabla H$ represent upslope flows. However, $(u_g - u_{gb})\cdot\nabla H$ represents the offshore geostrophic flow relative to the bottom (i.e. a thermal wind referenced to the bottom), which on an f-**

**plane has no associated dw/dz, and u is the total, depth-averaged flow, which includes the wind-driven Ekman flow - another component which need not involve any vertical motion. More insight comes if we write u = $u_g$ + $u_E$ , separating the depth-averaged flow into geostrophic and frictional (Ekman) components (the latter includes the effect of wind stress, bottom stress, and lateral friction). The important quantity can now be rewritten as -($u_E$ + $u_{gb}$)•∇H, representing the combination of Ekman and bottom geostrophic onshore flows. In shallow water for example, where we would expect the onshore wind-driven Ekman term to be increasingly balanced by offshore Ekman flow due to bottom (and lateral) friction, this term tends to zero as that balance is established, although the exchange of water between upper and lower Ekman layers represents a downwelling. Equally, a deep water balance of onshore wind-driven Ekman flow and offshore barotropic flow would clearly be a downwelling flow, but again this term would be zero. In short, it cannot meaningfully be described as an upwelling.**

Instead of describing the diagnostic as upwelling, we are now referring to it as net Ekman pumping velocity that results from the flow across isobars (l. 333). Following this modification, we have adjusted the wording throughout the text. Please let us know if we can bring this out more clearly.

**[R1.1.2] The discussion of the Stommel model seems irrelevant. All models are consistent in that the sea level slope at the boundary is related to the difference between wind stress and frictional stress at the boundary (this is simply the boundary condition), but beyond that the Stommel model depends essentially on beta - the boundary current represents a balance between bottom stress curl and the beta term - so the f-plane derivation of section 3 is not relevant.**

We have removed the discussion of the Stommel model from the manuscript.

**[R1.1.3] The Arrested Topographic Wave mode, while consistent with section 3, is barotropic and, in the light of the section 7 results which show the term related to stratification to be dominant, it seems to add little of relevance.**

The discussion of the idealized models serves primarily to illustrate the coastal and regional views, and, in conjunction with the realistic GoMSS model, show the versatility of these interpretations. In that regard, Section 5 (previously Section 4.2) is intended to provide more context to the theory in Section 4 rather than "explain" the results in Section 7. We have clarified the purpose of the section in the text:

*"In Section 4, two views of the dynamical role of the alongshore tilt of MDT at the coast are introduced and subsequently tested in both idealized models of shelf circulation (Section 5) and the realistic GoMSS model (Section 6)." (l. 64-66)*

*"In the following sections, we further discuss the coastal and regional views of Δη_c in the context of idealized models for shelf circulation and finally demonstrate that these interpretations also hold in the realistic, high-resolution model GoMSS. Overall, these sections serve to illustrate the versatility of the dynamical interpretations that further highlights the usefulness of the tilt of MDT for ocean model validation and monitoring." (l. 337-340)*

*"In Section 4 it was shown that the alongshore tilt of MDT at the coast can be interpreted in terms of the coastal and the regional circulation. Using idealized models, it was demonstrated that $\Delta\eta_c$ is a measure of the mean alongshore current at the coast (coastal view), but can also be related to the net Ekman pumping due to vortex stretching offshore (regional view). Here, we will test whether these views also hold in the realistic GoMSS model with a focus on the nearshore region between the reference points $s_1$ and $s_2$ outlined by the red polygon in Figure 1."* (l. 395-399)

**R1.2 Maths**

**The derivation of (22) is correct, but very roundabout, with a number of approximations introduced gradually through the derivation. It is in fact a slight rewrite of quite a standard equation (the use of the boundary condition being the main innovation). If we remove the nonlinear terms from (3) (these are incorrect because the depth average of, for example, u squared is not the square of depth-averaged u), and note that the term in brackets on the left is pb/g$\rho$0, replace h with H (an approximation used later in the paper), and introduce a streamfunction such that $\rho$0Hu = k x $\nabla\psi$, (3)xH becomes**

**(A) $f\nabla\psi = \rho_0\nabla\chi + H\nabla p_b - \tau$**

**($\tau$ represents all the friction terms).**

**Dividing (A)S by H then taking the curl gives the barotropic potential vorticity equation, which integrates to a form of (22) when the boundary condition is used to replace the wind stress integral (it is helpful to work in terms of depth-integrated pressure P instead of sea level at the boundary, noting that P = $\rho$0$\chi$ + Hpb). This provides a much more straightforward derivation, without the "upwelling" interpretation.**

We have simplified the derivations in Section 4 (previous Section 3) by stating all assumptions at the beginning, neglecting the inverse barometer effect, and starting from the linearized moment equation for depth-averaged flow using a slightly rewritten form of the one presented by Csanady (1979) with differences described in the text. The full definition of $\eta_s$ and $\chi$ are now given in (4) and (5). We also adopted your suggestion to suggestion go straight to H and the stress terms are now divided by $\rho_0$ (see comment R1.4.2).

**R1.3 Organization**

**It seems odd to have the derivation of (22) at the beginning of a paper which then focuses on the time-mean flow and the boundary interpretation. It would be much more helpful to have a self-contained "steady state, boundary interpretation" part of the paper, then move to the "regional, time dependent" ideas and diagnostics. I'm not sure how useful (22) actually is (it seems to be a way of assessing which part of the dynamics that needs ultimately to be balanced by a bottom pressure term, has not been balanced by it until the sidewall is reached, thus needing a pressure gradient (sea level slope) along the sidewall), but it is certainly interesting, and particularly interesting that the $\chi$ term plays such a big role. In many ways this is almost 2 separate**

**papers, but I do see the sense in keeping them together, as long as the logical progression is made clearer.**

We have restructured the manuscript by moving the model validation in Section 3 (previous Section 5) before the dynamical interpretation sections. This new structure divides the paper into two self-contained parts: the first part (Sections 2 and 3) addresses the first research question raised in the Introduction "can new observations of geodetically referenced coastal sea level help validate high-resolution models?" and the second part addresses the dynamical interpretation of the alongshore tilt of MDT using theory (Section 4), idealized models (Section 5), and the realistic numerical high-resolution regional model GoMSS (Section 6). Since the two interpretations complement each other, we prefer to present them side by side for each model.

**R1.4 Minor Issues**

**[R1.4.1] The description of how the permanent tide is accounted for is confusing (I sympathise! It is hard to explain this issue clearly). I suggest using (1) as the basis throughout, and explaining how he and N are calculated by correcting GPS heights and geopotential heights from tide-free to mean-tide system, and then applying (1), rather than saying (1) is applied then corrected.**

Following your suggestion, we first describe how he and N were calculated and then apply (1) to estimated coastal MDT. The modified text reads:

*"GPS coordinates are generally expressed in a tide-free coordinate system (Woodworth et al., 2012) as is the geoid model CGG2013a. In order to make geodetically referenced MSL observations comparable to ocean circulation models, mean tidal effects on the coordinate systems have to be considered. Therefore, he and N were calculated by converting GPS heights and geoid heights from tide-free to mean tide coordinates using the corrections provided by Ekman (1989). Note that the minus sign error reported by Woodworth et al. (2012) was taken into account." (l. 117-121)*

**[R1.4.2] Line 138 - note that MDT and model MSL differ by an unknown constant (dynamically irrelevant) offset.**

We have modified the text:

*"Therefore, MSL predicted by the model is equal to the MDT (plus an unknown, dynamically irrelevant constant) and can be directly compared to the geodetic estimates." (Line 137-138)*

**[R1.4.2] Equation 3 and line 168 - a full definition of χ is needed (I think it is the Mertz and Wright one, which is actually depth-integrated PE anomaly divided by $\rho_0$), the nonlinear terms should be removed (and I would recommend going straight to H instead of h), the wind stress term should be divided by $\rho_0$, and a reference should be given for the source of the equation (as noted above, there are other simplifications of presentation that could be made too).**

We have implemented your suggestions in the text. Please see response to comment R1.2 for details.

**[R1.4.2] Line 179 - this seems to be a definition of bottom pressure torque rather than the JEBAR term, which is better defined in the quotation used later.**

We have modified the text:

*"As shown by Mertz and Wright (1992), this can be interpreted as the baroclinic contribution to the torque related to the depth-averaged pressure." (l. 262-263)*

**[R1.4.2] Line 202 - "wind setup" suggests the effect of a wind blowing towards, not along the coast.**

We changed "wind setup" to variations of "sea level difference along the coast due to wind stress" and "wind-driven tilt along the coast" throughout the text.

**[R1.4.2] Line 254 - "corrected for" seems wrong here - u* is the total flow minus the geostrophic flow relative to that at the bottom.**

We have corrected the wording to

*"total flow minus the geostrophic flow relative to that at the bottom" (l.332)*

**[R1.4.2] Line 607-8 (regarding the scale factor) - but what would be an appropriate value to use for "water depth at the coast", when it is not in a model with a fixed, finite sidewall?**

*We now provide one approach how the water depth at the coast could be determined:*

*"For practical applications, when the water depth cannot be inferred from a numerical model with a sidewall, it could be determined from bathymetric soundings as the mean water depth along the outside the surf zone." (l. 557-559)*

**In conclusion, I see that the authors have done a lot of work to interpret the coastal sea level signals they are investigating. The data analysis is good, the topic and results are interesting if not completely conclusive, and the cited literature is appropriate - I would like to see this paper published. But it does need some streamlining and reorganising to make it clearer what has actually been shown, and to improve the logical flow of the ideas.**

Thank you again for your constructive feedback! We hope the restructured manuscript and clarifications outline above have helped improve the logical flow of the ideas and make the paper easier to follow.

**Reviewer 2**

**This is an interesting extension of earlier studies comparing the geodetic and ocean model approaches to the estimation of the alongshore tilt of MDT. The paper is broken into two main sections – the first looking at the geodetic estimates to validate a regional model, and the second considering how the alongshore slope can provide information on the shelf circulation.**

**Overall the first section, using geodetic estimates of MDT to validate the GoMSS regional model, is a worthwhile addition to the literature. However I'm left with the - sense that spatially-sparse geodetic measurements with relatively high uncertainties are of limited utility in validating a high-resolution model such as GoMSS. I think that the authors slightly overstate the agreement between the model and observations. Qualitatively there is broad agreement, but only when some measurements are excluded and other discrepancies explained.**

**The second section, deriving and describing the relationship between the alongshore tilt of MDT and the coastal and regional circulation, is less compelling. The coastal view, as it is described by the authors, is similar to the momentum balance presented by Lin et al., but improved through the use of a higher-resolution model. This is of value in terms of understanding the tilt of MDT and, perhaps, in terms of monitoring the alongshore flow. The value of the regional view is less clear.**

**The layout of the manuscript is challenging. Whilst I think I understand why the derivations are presented up front (sections 3 and 4), these sections are lengthy and heavy going. The results and interpretation sections consequently appear somewhat lost.**

Many thanks for your valuable comments and suggestions on our paper! We addressed your concerns as described below and hope that the revised manuscript is now easier to follow.

**I suggest the following to improve the manuscript:**

**[R2.1] Consider reducing the derivations and/or moving them to an appendix. Whilst I won't try to fault the math, the derivations are convoluted and rely on many assumptions along the way. Much of this is published elsewhere and I can't help but think that it could be simplified.**

We have reduced and simplified the derivations in Section 4 (previous Section 3) by stating all assumptions at the beginning, neglecting the inverse barometer effect, and starting from the linearized moment equation for depth-averaged flow using a slightly rewritten form of the one presented by Csanady (1979) with differences described in the text (see response to comment R1.2 for details). We have also removed the discussion of the Stommel model (previous Section 4.1).

**[R2.2] I'm not convinced that the regional view is adding to the story here. It seems more of a theoretical study that doesn't sit well with the rest of the paper. If it is to remain, the utility of an integrated measure of upwelling needs to be demonstrated.**

The usefulness of $\Delta\eta_c$ for model validation and implications ocean monitoring are discussed in the final paragraph of Section 7. We modified the text to highlight the value for model validation, as shown in our study. The implications for ocean monitoring are intentionally speculative and provide directions for future studies:

*"This close relationship between coastal MDT and integrated nearshore circulation highlights utility and value of tilt estimates based on geodically referenced sea level observations for model validation. Furthermore, it also has obvious implications for ocean monitoring: For example it may be possible to use long coastal sea level records to estimate time series of upwelling rates. Such information may be of interest to biological oceanographers interested in understanding changes in nutrient cycling on the shelf over recent decades. This speculation applies not only to the Scotian Shelf. For example, in the future, it would be interesting to test the idea on the west coast of North America given the large number of long, geodetically referenced sea level records (e.g., Lin et al., 2015) and the large amount of hydrographic data, e.g., the California Cooperative Oceanic Fisheries (CalCOFi) Database. Finally, our findings have implications for future deployments of tide gauges if they are to be used to monitor the shelf-scale ocean circulation. Coastal MDT can be affected by local processes, e.g., strong tidal flow around headlands and therefore, the location of the tide gauges should be exposed to the open ocean and at distance to areas where local processes dominate."* (l. 575-585)

**[R2.3] The final paragraph of section 5 (lines 446 to 451) would benefit from some edits. The "good agreement" is qualitative and subject to a number of caveats (for example, the removal of some stations, and the uncertainty in the geodetic measurements). Also I think that the statement that "The agreement gives confidence that GoMSS captures the mean circulation, including the effect of tidal rectification…" is a little misleading. I believe that the set down at Yarmouth is the evidence of tidal rectification, but I struggle to see anything in the geodetic measurements to really support this.**

We have reworded that part of the text that now reads:

*"The overall agreement of $\Delta\eta_c$ estimated independently by the hydrodynamic and geodetic approaches provides validation of the ocean model. This gives confidence that GoMSS captures the mean shelf-scale ocean circulation on the Scotian Shelf and in the Gulf of Maine. While there are some limitations to this method (see Section 7), the dynamical interpretation of $\Delta\eta_c$ below illustrates that the alongshore tilt of MDT is an integrated measure of nearshore ocean dynamics highlighting its usefulness for model validation. Specifically, the next two sections, the address the following questions: What can the alongshore tilt of MDT at the coast tell us about shelf circulation? What are the implications for coastal monitoring?"* (Lines 235-241)

Additionally, we have added the following text to the discussion in Section 7:

*"It is important to note that some tide gauges are deployed in geographic settings (e.g., inside harbors or behind sandbanks) that are not resolved by ocean models and where sea level variations are likely dominated by local processes. Therefore, a comparison between observed and predicted MDT is only meaningful in locations that are relatively exposed to the open ocean. At the tide gauges considered in this study, the uncertainties in the geodetic MDT estimates are less than 1.6 cm and are primarily due to estimated error of the geoid*

*model. With the ongoing efforts to improve these models, the uncertainties are expected to become smaller in the future. Nevertheless, the use of $\Delta\eta_c$ for model validation is limited to regions with long, geodetically referenced sea level records."* (l. 535-541)

**In conclusion I think that the paper is interesting and worthy of publication. However I think that it needs some significant reformatting to improve the readability, and perhaps a reduction in scope to focus on the key messages.**

Thanks again for your feedback! Your comments are much appreciated.

**References**

Csanady, G T. 1979. "The Pressure Field along the Western Margin of the North Atlantic." *Journal of Geophysical Research* 84 (C8): 4905. https://doi.org/10.1029/JC084iC08p04905.

Mertz, Gordon, and Daniel G. Wright. 1992. "Interpretations of the JEBAR Term." *Journal of Physical Oceanography* 22 (3): 301–5. https://doi.org/10.1175/1520-0485(1992)022<0301:IOTJT>2.0.CO;2.

---

## Author Response (AR2)

**The Alongshore Tilt of Mean Dynamic Topography and its Implications for Model Validation and Ocean Monitoring**

Christoph Renkl, Eric C.J. Oliver and Keith R. Thompson

**Response to Comments by Reviewers**

We thank the reviewer again for their insightful comments – we have responded to all of them as detailed below. The reviewers' comments are repeated in bold font, and our responses are given in regular font with modified parts of the text, including the respective line numbers, by italicized font. We believe the changes have resulted in a significant improvement to the manuscript and hope the reviewer and editor will find the revised version suitable for publication.

**Reviewer 1**

**The authors have made substantial changes to the manuscript. The mathematical derivations are now more succinct, and the rearrangement makes the logic of the arguments much clearer. I would like to thank them for these clarifications and responses to my (and the other reviewer's) earlier comments. The paper is greatly improved.**

Thank you again for your very insightful comments and constructive feedback on our manuscript! Following your suggestions, we have changed the wording around the interpretations and addressed your comments as detailed below.

**However, I do still have some significant problems with the interpretation of the "regional view" analysis, which I feel really does still need some further changes before it can be considered ready for publication. There is also one issue with the coastal interpretation, and some other minor issues which I list below.**

**The maths of the regional view is correct, and the analysis is interesting. In particular, it is intriguing that the JEBAR term appears so dominant in this analysis, in both mean state and time dependence, when the coastal balance is so dominated by the winds. I am persuaded that the presentation of the Csanady model is worthwhile for the insight it gives into both wind forced and "downstream" responses for the barotropic case, as a contrast to what is actually seen in the model.**

**The problem is with the interpretation as "upwelling", and the descriptions in various places referring to vortex stretching. I have tried for quite some time to find a good physical interpretation of the balance that has been presented, and have not come up with anything clearer, but what is clear is that w* is not related to upwelling in any meaningful sense.**

**To make this clear, consider a few examples.**

1) Consider a wind stress which does not intersect the coast, and has no curl. This will drive a recirculating Ekman flux with no divergence, and so the only flow will be this Ekman flux. From (19), u* is then the Ekman flux divided by H, and w* from (20) can be positive or negative, or zero, depending on how the topography is orientated relative to the Ekman flux. In this scenario there is no upwelling or downwelling. There will also be no coastal sea level signal. So w* can be nonzero when w is zero.

2) Close to the coast in the diagnostics presented here, the primary balance is between wind stress (an Ekman flux away from the coast) and pressure gradient (a geostrophic flow towards the coast), with bottom stress a minor contributor in most places, and no depth-average flow toward or away from the coast. It is also close to barotropic, so geostrophic u and bottom geostrophic u are equal in (19) making u* a flow along the coast (along isobaths), and therefore w* is zero from (20). However, this represents an upwelling flow as the geostrophic flow toward the coast is at greater depth than the Ekman flux away. So w can be zero in an upwelling flow.

3) In a region where the flow does not reach the seafloor (so the geostrophic component can be computed by thermal wind relative to the bottom), and a wind-driven Ekman flow away from the coast is balanced by a geostrophic flow towards the coast, the depth-averaged u and bottom geostrophic u in (19) would both be zero, so u* would be minus the thermal wind flow, making w* -ve from (20). But in this region there is no vertical velocity (no convergence of the Ekman flux), and the oceanward surface Ekman flux balanced by a coastward geostrophic flow at greater average depth actually implies an upwelling somewhere coastward of this region. So w* can be nonzero when w is zero, and the flow is not confined to the surface Ekman layer.

4) The diagnostics shown in Fig. 8 illustrate a case in which the JEBAR term (coming from geostrophic u minus bottom geostrophic u in (19), or thermal wind flow relative to the bottom) dominates, and is larger than the depth-averaged flow effect. In that case, from (19) and (20), w* is negative as observed if the thermal wind flow is towards the coast. With the wind-driven Ekman flow being away from the coast, this is clearly an upwelling scenario similar to 3) above.

5) In the Csanady model the JEBAR term is absent. In this case, it is only the depth-averaged flow that matters in (19), and w* can be diagnosed from the streamlines in Fig. 5, being positive in the upper part of the figure where the depth-integrated flow is toward the coast and negative in the lower part of the picture. But there is no wind stress curl in this configuration, so w is zero near the surface, and a downslope geostrophic flow requires vortex stretching, and hence w negative through the interior of the flow where the geostrophic flow is downslope, and vice versa. The right hand panel shows that this geostrophic flow is (in the sense of towards or away from the coast) in the opposite direction to the depth-integrated flow in the northern region and in much of the southern region, so the actual w in a large part of the flow has the opposite sign to w*. On the other hand, with southward geostrophic currents everywhere, the bottom Ekman flux will be away from the coast everywhere and thus w will be negative in the bottom Ekman layer. If this layer has sufficient weight in the chosen form of vertical average, it will make w* have the same sign as vertically averaged w in the southern region, but will exacerbate the mismatch in the north. So a full solution can have different relationships between w* and w in different regions.

**From the above examples it should be clear that there is no meaningful sense in which positive w\* can be held to represent upwelling. It seems more likely to be a downwelling, but is not in general related to the vertical velocities in any consistent way. In the same way, w\* has no consistent relationship to Ekman pumping or vortex stretching (see example 1 above), and interpreting the curl(tau/H) term as Ekman pumping is incorrect, since it includes both wind stress curl and a term tau x grad(H) which represents the Ekman flux across depth contours, which need not be associated with any vertical velocity.**

**It is unfortunate that there seems to be no simple interpretation of u\*.grad(H), but the results associated with it do seem very interesting. In particular, Fig. 8 shows that the term which is absent in the Csanady model is actually dominant in this simulation, which is interesting for such a shallow region (perhaps a measure of the strong freshwater influence from river input?). It is a particularly striking contrast to Lentz (2024). I suggest removing any mention of upwelling, and removing the w\* notation which misleads through its implied relationship to w. Other, similar phrases such as "area integrated vortex tube stretching" (line 489) and "associated net Ekman pumping velocity" (line 333, see also 397) should also be removed as they are inaccurate. Similarly, the introduction of xi (line 495) is unhelpful, and the diagnostics in Fig. 8 should be presented in terms of sea level change, to remain consistent with other diagnostics, and to ensure that the scale is meaningful.**

**I would like to see these results kept in the paper because they are very interesting, even if a clear interpretation cannot be reached. But it is important that the misleading interpretations be removed.**

Thank you for your detailed examples – we recognize that, unfortunately, there is no simple interpretation of $u^* \cdot \nabla H$, in particular with regards to upwelling. Your efforts to find a good physical interpretation are much appreciated! We have addressed your concerns by making the following changes in how the results are framed in the manuscript:

Based on your suggestions, we have removed the definition of w\* (previous Eq. (20)). We furthermore removed any links between the regional view and upwelling in the interpretation of the GoMSS output and also removed any inaccurate phrases mentioned in your comments. Instead, we now use the phrase "the tilt provides a measure of area-integrated nearshore circulation" and variations thereof.

Additionally, we removed the definitions of $\xi$ (previous Eqs. (33) and (34)) and express the diagnostics shown in Figure 8b in terms of $\Delta\eta_c$.

**Another small but important issue is in the interpretation of the coastal balance, starting around line 425. Here, and elsewhere, it is stated that bottom friction partly balances wind stress. In fact, Fig. 7 shows that it acts in concert with wind stress in most regions. This is consistent with the fact that the Nova Scotia Current flows in the opposite direction to the wind stress.**

As suggested, we modified the text to reflect the fact that the alongshore tilts due to wind stress and bottom friction have the same sign along most of the coast line of Nova Scotia. The new text reads:

*"Along most of the coastline, this wind-driven tilt acts in concert with bottom friction ($\Delta\eta_{BF}$) acting on the Nova Scotia Current flowing in the opposite direction of the wind stress at the coast. An exception to this is the region around Sydney where bottom drag is strongest and associated with the Nova Scotia Current flowing close to the coast as it exits the Gulf of Saint Lawrence." (l. 427)*

**Minor points**

**Line 35: In fact Huang only says the accuracy can be at the centimetre level for low-lying coastal regions, and errors can be decimetric on mountainous coasts. It would be worth qualifying this statement.**

We added a qualifying statement and the updated text reads:

*"In coastal regions with flat topography, these independent measurements are accurate on the centimeter level (Huang 2017) and thus provide potentially valuable information for the validation of ocean and shelf circulation models." (l. 35)*

**Lines 265 and 353: As above, the curl(tau/H) term is a combination of a "torque" (in the same sense as used in the term bottom pressure torque) and an Ekman flux across depth contours.**

*Changed to:*

*"The remaining terms in (6) describe the curl of the difference between wind stress and bottom friction and their associated Ekman transports across depth contours, as well as vorticity dissipation due to lateral mixing." (l. 267)*

*"The combination of the net torque exerted by the wind stress and bottom drag (right-hand side) and their associated Ekman transports across depth contours, is balanced by vortex stretching/squashing through movement into deeper or shallower water, respectively." (l. 354)*

**Line 379: "Convergence of this onshore flow implies downwelling that is balanced by return flow in the frictional bottom boundary layer" is confusing. The wind-driven Ekman flux at the coast implies downwelling but this is only partially balanced by a frictional return flow. Some of the return flow is geostrophic, and some does not return but flows along the coast. The total onshore flow (which is what matters for u* in this model) is not balanced by any offshore flow by definition, but by a divergence of the alongshore flow. Please clarify.**

We clarified this statement and the new text reads:

*"The onshore wind-driven Ekman transport implies downwelling at the coast that is partially balanced by a return flow in the frictional bottom boundary layer and partially by an offshore geostrophic flow in the interior. The total onshore flow is balanced by a divergence of the*

*alongshore flow. The rate of water exchange between the surface Ekman layer and the interior of the water column in that area can be monitored by observing the sea level at the coast." (l. 381)*

**Line 437-8: It is only the boundary condition of the Csanady model that is justified by these diagnostics, not its application in the interior.**

We added the qualifying statement *"as coastal boundary" (l. 441).*

**Line 502: typo "with and"**

This sentence has been removed.

**Line 518-9: I don't follow this statement. Is it in relation to the curl(tau/H) term again? If so, it is a misinterpretation as the vorticity input is only due to curl(tau).**

We clarified this statement. The new text reads:

*"Although the wind stress can be uniform over a large area, the associated Ekman transport toward the deeper region offshore leads to a change in relative vorticity." (l. 512)*

**Line 554: please change "wind setup" to "coastal sea level slope due to alongshore wind stress"**

We changed "wind setup" to *"tilt due to alongshore wind stress" (l. 547)*